# Improved Localized Machine Unlearning Through the Lens of Memorization

## Abstract

Machine unlearning refers to removing the influence of a specified subset of training data from a machine learning model, efficiently, after it has already been trained. This is important for key applications, including making the model more accurate by removing outdated, mislabeled, or poisoned data. In this work, we study localized unlearning, where the unlearning algorithm operates on a (small) identified subset of parameters. Drawing inspiration from the memorization literature, we propose an improved localization strategy that yields strong results when paired with existing unlearning algorithms. We also propose a new unlearning algorithm, Deletion by Example Localization (DEL), that resets the parameters deemed-to-be most critical according to our localization strategy, and then finetunes them. Our extensive experiments on different datasets, forget sets and metrics reveal that DEL sets a new state-of-the-art for unlearning metrics, against both localized and full-parameter methods, while modifying a small subset of parameters, and outperforms the state-of-the-art localized unlearning in terms of test accuracy too.

## 1 Introduction

Machine unlearning, coined by Cao & Yang (2015), is the problem of removing from a trained model (the influence of) a subset of its original training dataset. While unlearning is a young area of research, it has recently attracted a lot of attention (Triantafillou et al., 2024). Example applications of unlearning include keeping models up-to-date or improving their quality by deleting training data that is identified post-training as being outdated, mislabeled or poisoned.

Unlearning is a challenging problem in deep neural networks since they are highly non-convex, preventing us from easily quantifying the influence of different training examples on the trained weights. As a straightforward solution to unlearning a given "forget set", one can simply retrain the model from scratch on an adjusted training dataset that excludes that set. This approach implements *exact unlearning*, guaranteeing that the resulting model has no influence from the forget set. However, this approach can be prohibitively computationally expensive. Instead, a burgeoning area of research has emerged that designs methods to post-process the trained model to attempt to *approximately* erase the influence of the forget set, *efficiently*. This post-processing introduces a challenging balancing act, as imperfect attempts at removing some training examples after-the-fact may accidentally damage the model and overly reduce its utility (e.g. accuracy on the remainder of the training data or generalization ability). Therefore, designing successful approximate unlearning methods involves navigating trade-offs between i) forgetting as well as possible, ii) utility, and iii) efficiency.

We hypothesize that *localized unlearning*, where the unlearning algorithm operates on only a (small) subset of the parameters, is a promising avenue for striking a good balance in the above trade-offs. Specifically, modifying a (appropriately chosen) small fraction of the weights may be intuitively less likely to overly damage the network's utility (e.g. generalization capabilities) and more likely to be efficient, since fewer parameters are subject to modification. However, the success of such a localized approach hinges on the ability to identify the right subset of parameters to perform unlearning on. In this work, we take a deep dive into different localization strategies, drawing inspiration specifically from hypotheses formulated about *where* in the network training data is "memorized" [1].

---

[1] We intend here a very restricted definition of "memorization" in the context of classification models, which do not provide generative outputs. In this context, memorization relates to a data element contributing to the

Indeed, a closely-related research community has been studying *memorization* in neural networks. Informally, a training example is memorized by a model if that model's predictions on that example would have been different had the example not been included in the training set (Feldman, 2020; Pruthi et al., 2020). As we discuss later, this notion is closely tied to unlearning. In this work, we investigate whether hypotheses for where memorization happens give rise to improved localized unlearning, through informing which parts of the network we should act on in order to unlearn a given set of examples. Our contributions can be summarized as follows:

- We leverage hypotheses for where memorization occurs to derive strategies that pinpoint a minimal set of parameters that the unlearning algorithm should act on; and we investigate their strengths and weaknesses. We find that data-agnostic strategies are a poor choice: they either achieve good unlearning performance at the expense of utility, or the other way around, but not both. We aim to improve on this via data-dependent approaches that target a small fraction of the parameters chosen based on the particular forget set.

- Our insights from a thorough investigation led us to propose a practical localization strategy inspired by the memorization localization algorithm of Maini et al. (2023) that is more efficient than that algorithm and, when paired with various unlearning algorithms from the literature, outperforms prior work in terms of unlearning and utility metrics.

- We propose a new localized unlearning algorithm, Deletion by Example Localization (DEL), by pairing our localization strategy with the simple approach of resetting the deemed-to-be critical parameters and then finetuning the newly reinitialized parameters.

- DEL outperforms the state-of-the-art localized and full-parameter methods on unlearning efficacy, and all localized methods on utility too, on different forget sets and datasets. Unlike other strategies, DEL can achieve strong results for several different parameter budgets.

## 2 BACKGROUND

We begin by introducing the notation we will use in this paper and defining key concepts.

Let $\mathcal{D}_{\text{train}}$ denote a training dataset and $\mathcal{A}$ a (possibly randomized) training algorithm. Then, we denote by $\theta^o = \mathcal{A}(\mathcal{D}_{\text{train}})$ the parameters obtained by training on $\mathcal{D}_{\text{train}}$ using $\mathcal{A}$. We will refer to $\theta^o$ as the "original model", i.e. the model before any unlearning takes place. We will study algorithms for "unlearning" a subset $\mathcal{S} \subset \mathcal{D}_{\text{train}}$, referred to as the forget set. We refer to $\mathcal{D}_{\text{train}} \setminus \mathcal{S}$, the remainder of the training data, as the retain set. While different variations are possible, we assume for simplicity that the unlearning algorithm $\mathcal{U}$ has access to both the forget set and the retain set.

### 2.1 UNLEARNING

In this section, we define unlearning intuitively in a way that faithfully reflects the standard metrics used in the community that we also adopt for evaluation; see Section A.1 for an alternative definition.

**Definition 2.1. Unlearning.** For a given algorithm $\mathcal{A}$ and dataset $\mathcal{D}_{\text{train}}$, an algorithm $\mathcal{U}$ is said to unlearn a forget set $\mathcal{S}$ if the unlearned model $\mathcal{U}(\theta^o, \mathcal{S}, \mathcal{D}_{\text{train}} \setminus \mathcal{S})$ and the "retrained model" $\mathcal{A}(\mathcal{D}_{\text{train}} \setminus \mathcal{S})$ have the same distribution of outputs on $\mathcal{S}$.

The above compares the (distribution of) outputs of the models obtained by two different recipes. The first is $\mathcal{A}(\mathcal{D}_{\text{train}} \setminus \mathcal{S})$, retraining "from scratch" on only the retain set, which is prohibitively expensive but ideal from the standpoint of eliminating the influence of $\mathcal{S}$ on the model. The second is $\mathcal{U}(\theta^o, \mathcal{S}, \mathcal{D}_{\text{train}} \setminus \mathcal{S})$, applying $\mathcal{U}$ to post-process the original model $\theta^o$ in order to unlearn $\mathcal{S}$.

In the above, we refer to "distributions" of outputs since re-running either of the two recipes with a different random seed, that controls the initialization or the order of mini-batches, for example, would yield slightly different model weights, thus possibly slightly different outputs too. In our experiments considering classification tasks, the "outputs" are the vector of softmax probabilities, and different metrics consider different elements of that vector, e.g. the accuracy metric requires the "argmax" of that vector, whereas more sophisticated metrics consider the correct class probability ("confidence").

---

model's ability to accurately label input data. Such models do not "contain" bit-wise or code-wise copies of their training data.

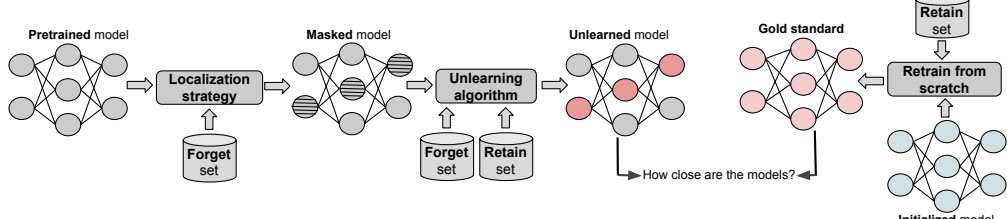

Figure 1: **Localized unlearning** consists of two parts: a *localization strategy* that identifies a set of "critical parameters" (dashed line circles) and an *unlearning algorithm* that aims to remove the influence of the forget set by modifying only the critical parameters (highlighted circles), keeping the rest unchanged. Ideally, the unlearned model should "behave" like the model retrained from scratch, i.e. the two should produce the same (distribution of) outputs; see Definition 2.1

We desire unlearning algorithms $\mathcal{U}$ that cause these two recipes to yield similar outputs, with the second recipe being substantially more computationally-efficient compared to the first, in order to justify paying the cost of approximate unlearning rather than simply using the first recipe directly.

**Unlearning evaluation.** Evaluating unlearning rigorously is an ongoing area of research; current state-of-the-art evaluation methods (Hayes et al., 2024; Triantafillou et al., 2024) require training a large number of models, which is very expensive. In this work, we leverage standard metrics in the research community, building on top of the evaluation procedure of (Fan et al., 2023) that considers two metrics for unlearning quality. The first is the accuracy of the unlearned model on the forget set, with the goal of matching the accuracy of the retrained model on the forget set, in line with Definition 2.1. The second is a Membership Inference Attack (MIA), that, given access to outputs ("confidences") of the unlearned model, aims to detect whether an example was used in training. We adopt the $MIA_{\text{efficacy}}$ score of Fan et al. (2023) that measures the efficacy of defending such an attack as the portion of forget set examples that the attacker thinks were unseen. An ideal unlearning algorithm would have an $MIA_{\text{efficacy}}$ score matching that of the retrained-from-scratch model; see Section A.3 for details. In addition, a comprehensive evaluation of unlearning also requires measuring utility, which we measure via test accuracy (and retain accuracy, in Section A.8), and efficiency.

**Localized unlearning.** In this work, we focus on "localized unlearning" algorithms $\mathcal{U}$ that modify only a (preferably small) subset of the parameters of $\theta^o$, leaving the rest unchanged (see Figure 1). We view localized unlearning as a promising direction as we hypothesize that it can yield better trade-offs between unlearning efficacy, utility and efficiency, due to modifying fewer parameters.

A localized unlearning algorithm has two components. The first is a *localization strategy* $\mathcal{L}$ that produces a mask $m$ determining which subset of the parameters should be modified to carry out the unlearning request: $m = \mathcal{L}(\theta^o, \mathcal{S})$ where $m$ is a binary vector specifying whether each parameter will be updated. The second component is a *unlearning strategy*. This, in principle, can be any unlearning algorithm that, in this case, will operate on only the subset of the parameters indexed by $m$. Overall, a localized unlearning algorithm is instantiated by selecting a particular $(\mathcal{L}, \mathcal{U})$ pair.

## 2.2 MEMORIZATION

An intriguing phenomenon is that, despite models exhibiting strong generalization properties, they still tend to "memorize" some of their training data (Arpit et al., 2017; Zhang et al., 2021). In fact, recent theories argue that some forms of memorization are in fact necessary for optimal generalization (Feldman, 2020; Brown et al., 2021; Attias et al., 2024). In the below, we first present a definition of memorization borrowed from (Feldman, 2020), and then discuss the connections with unlearning.

**Definition 2.2. Label memorization.** Assume a dataset $\mathcal{D}_{\text{train}} = \{(x_i, y_i)\}_{i=1}^N$, where $x_i$ and $y_i$ denote the input and label for the $i$'th training example, and assume a training algorithm $\mathcal{A}$ and a model $f(x; \theta)$ parameterized by $\theta$ mapping inputs to labels. Then, the *memorization score* for an example $(x_i, y_i) \in \mathcal{D}_{\text{train}}$ (with respect to $\mathcal{D}_{\text{train}}$, $\mathcal{A}$ and $f$) is

$$\Pr_{\theta \sim \mathcal{A}(\mathcal{D}_{\text{train}})}[f(x_i; \theta) = y_i] - \Pr_{\theta \sim \mathcal{A}(\mathcal{D}_{\text{train}} \setminus (x_i, y_i))}[f(x_i; \theta) = y_i]. \quad (1)$$

Intuitively, an example is highly memorized by a model if the model can only predict its label correctly when that example is in the training set. This will be primarily the case for atypical, ambiguous or mislabeled examples that would not be otherwise correctly predicted (Feldman & Zhang, 2020).

**Connections with unlearning.** Some forms of memorization and unlearning are intimately connected: at the extreme where an example isn't memorized at all, it can be thought of as being trivially "unlearned" according to some unlearning metrics of interest, because the model's predictions on that example aren't different from what they would have been had that example not been included in the training set ("retrain from scratch"). Empirically, considering varying degrees of memorization, Zhao et al. (2024) showed that most approximate unlearning methods are more successful on forget sets which include examples that have lower memorization scores compared to those with higher memorization scores. Relatedly, Jagielski et al. (2022) study catastrophic forgetting during training; a phenomenon that can be characterized as reduced memorization of an example in later stages of training, that can be interpreted as a passive form of unlearning. They find that, when training on large datasets, examples that were only seen early in training may be less memorized, which they quantify via the failure rates of privacy attacks aiming to extract examples or infer whether they were used for training. Toneva et al. (2018) find that examples with noisy labels witness a larger number of "forgetting events" during training, defined as an event where an example that was previously correctly predicted becomes incorrectly predicted later in training. Based on these insights on the strong connection of memorization and unlearning, we ask: does knowledge (or assumptions) of *where* in the network a forget set is memorized give rise to improved unlearning for that forget set?

**Localizing memorization.** Investigating the above question is a challenging undertaking, as pinpointing where memorization occurs is in and of itself a research problem. Baldock et al. (2021) define the "prediction depth" for an example to be the earliest layer in the network after which the example is correctly predicted. They find that mislabeled examples are only predicted correctly in the final few layers of the model, and conclude that "early layers generalize while later layers memorize". Stephenson et al. (2021) draw the same conclusion through a study of manifold complexity and shattering capability. However, Maini et al. (2023) found that the parameters that memorize specific examples are actually scattered throughout the network and not concentrated in any individual layer.

## 3 RELATED WORK

**Unlearning.** Cao & Yang (2015) coined the term unlearning and proposed exact algorithms for statistical query learning. Bourtoule et al. (2021); Yan et al. (2022) propose frameworks that support exact unlearning in deep networks more efficiently by considering architectures with many components, where one only needs to retrain the affected parts of the model for each unlearning request. However, in the worst case, efficiency can be as poor as in naive retraining and, furthermore, these specialized architectures may have lower accuracy compared to state-of-the-art ones. Instead, a plethora of approximate unlearning methods (Ginart et al., 2019; Guo et al., 2019; Golatkar et al., 2020a;b; Thudi et al., 2022) were developed that operate on an already-trained model to remove the influence of the forget set. Simple commonly-used baselines include finetuning the model on only the retain set ("**Finetune**"), or on only the forget set using the negated gradient ("**NegGrad**"), or combining these two in a joint optimization "**NegGrad+**" (Kurmanji et al., 2024), performing gradient descent on the retain set and ascent on the forget set, simultaneously. One could also apply a joint optimization using gradient descent on both the retain and forget sets, after having first randomly relabelled the examples in the forget set ("**Random Label**") (Graves et al., 2021; Fan et al., 2023). **SCRUB** (Kurmanji et al., 2024) builds on NegGrad+ by casting unlearning as a teacher-student problem and using distillation. Liu et al. (2024) show that sparsity aids unlearning and that adding an L1-penalty to the Finetune baseline improves its performance ("**L1-sparse**"). One could also utilize influence function analysis (Koh & Liang, 2017) to mitigate the impact of the forget data by estimating the importance of the model weights ("**IU**") (Izzo et al., 2021).

**Localized unlearning.** Goel et al. (2022) propose baselines that perform unlearning on only the $k$ deepest (closest to the output) layers and either simply finetune them ("**CF-k**") or reinitialize them and then finetune them ("**EU-k**"). The contemporaneous work of Foster et al. (2024) ("**SSD**") uses the Fisher information matrix to identify parameters that are disproportionately important to the forget set and apply unlearning on those. Fan et al. (2023) proposes Saliency Unlearning ("**SalUn**"), which selects a subset of critical parameters by considering the gradients of the forget set and applies unlearning (by default using Random Label) on the identified parameters. We will later compare this state-of-the-art method to other strategies inspired by the memorization literature, and propose a novel strategy that outperforms previous work. Related ideas have been proposed in the context of unlearning in LLMs (Hase et al., 2024; Guo et al.), though we note that the problem formulation of LLM unlearning as well as the architectures and algorithms used there are substantially different and

out of the scope of this paper. We will later discuss how our findings relate and differ from theirs (See Section 7).

## 4 DERIVING AND INVESTIGATING LOCALIZED UNLEARNING BASED ON MEMORIZATION HYPOTHESES

In this section, we derive localization strategies from hypotheses in the memorization literature and investigate their performance in the context of localized unlearning compared to existing approaches. Based on the discussion in Section 2.2, we consider two hypotheses for where memorization occurs, and we then turn each one of them into a localization strategy $\mathcal{L}(\theta^o, \mathcal{S})$. The two hypotheses are: i) memorization happens in the "deepest layers", i.e. closest to the output, consistent with (Baldock et al., 2021; Stephenson et al., 2021), and ii) memorization of an example is confined to a small set of channels, scattered across the network, whose location depends on the example (Maini et al., 2023).

We compare the unlearning performance of four localization strategies: i) **Deepest** and ii) **CritMem**, derived based on the above two hypotheses, iii) selecting the shallowest rather than deepest layers ("**Shallowest**"), as a control experiment, and iv) the localization strategy of SalUn (Fan et al., 2023) ("**SalLoc**") that is not motivated through the lens of memorization but is regarded as state-of-the-art in the unlearning literature. Note that we reserve the name "SalUn" for the pairing of SalLoc with the Random Label unlearning algorithm, which is the choice that Fan et al. (2023) found worked best (we mix-and-match localization strategies with unlearning algorithms in Section 5).

**Deepest** returns a mask that allows only the $k$ deepest (closest to output) layers to be updated during unlearning, based on the first hypothesis. If paired with an unlearning algorithm that resets the chosen parameters and then finetunes them, this corresponds to the EU-k method of (Goel et al., 2022).

**CritMem** is based on the second hypothesis and implemented via the algorithm of Maini et al. (2023). Given an example, it iteratively finds the deemed-to-be most critical channel, resets all its associated parameters and repeats, until the prediction on that example flips to an incorrect one. The criterion used to identify the next most critical channel at each iteration is obtained by multiplying the weights by their respective gradients on that example, and choosing the channel for which the (sum, over the channel parameters) of the magnitude of this quantity is the highest. We run this algorithm for each example in $\mathcal{S}$ independently, record the critical channels identified for each and then consider the union of those as the critical channels for $\mathcal{S}$. The mask is then set to allow to only modify the parameters involved in those critical channels.

**Shallowest** returns a mask that allows only the $k$ shallowest (closest to the input) layers to be updated during unlearning, as a control experiment for Deepest.

**SalLoc** sorts all parameters based on the magnitude of gradients over $\mathcal{S}$, in descending order and then, given a threshold $\alpha$ for the percentage of the parameters that might be updated during unlearning, it returns a mask that turns on only the first $\alpha$ percent of elements of that sorted list (Fan et al., 2023).

The above strategies differ along several axes. Firstly, Deepest and Shallowest are data-agnostic in that they do not use $\mathcal{S}$ to inform which parameters to choose, whereas CritMem and SalLoc are data-dependent and tailor the mask to $\mathcal{S}$. Secondly, there are differences in granularity: Deepest and Shallowest consider a layer as the unit (i.e. each layer is either included or excluded as a whole). CritMem's unit is a channel while SalLoc has the finest granularity, considering each individual parameter as a unit. Finally, out of the data-dependent ones, CritMem is substantially more computationally expensive than SalLoc. This is because *for each* example in $\mathcal{S}$, it applies an *iterative* approach that resets the next most critical channel, one at a time, until the termination criterion. On the other hand, SalLoc chooses the critical parameters in "one-shot" rather than iteratively and, additionally, operates on *batches* of $\mathcal{S}$. Both of these aspects make it significantly more efficient.

**Experimental setup** We conduct this investigation on CIFAR-10 using ResNet-18, and a forget set comprised of $10\%$ of the training samples, randomly selected from two classes (2 and 5). We use a simple unlearning algorithm that resets the identified critical parameters (obtained based on each localization strategy), and then finetunes only the identified critical parameters and the classifier layer using only the retain set ("**Reset + Finetune**"). Since each of the above localization strategies chooses a different set and number of parameters, to ensure fairness, we compare them to one another for different "budgets" of how many parameters are updated. For example, for Deepest, we select the

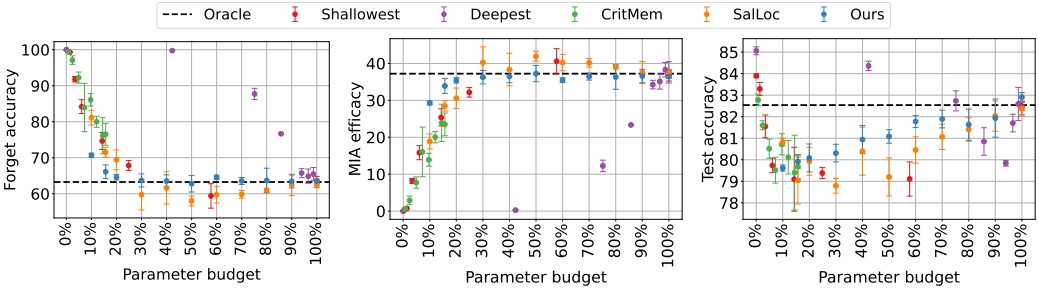

Figure 2: Comparison of localization strategies combined with the Reset + Finetune (RFT) unlearning algorithm. An ideal unlearning algorithm would match the "oracle" ("retrain-from-scratch") on each metric, with the smallest possible parameter budget, for increased efficiency. The strategy we will propose later ("ours") yields the best trade-off, with near-perfect unlearning for several budgets.

$k$ deepest layers (for several different values of $k$), and we compute the number of parameters that are selected each time to obtain the "budget". We measure unlearning performance in terms of three metrics: Forget accuracy and MIA efficacy for unlearning quality, and test accuracy for utility. For all three metrics, the ideal behaviour is to match the performance of the retrain-from-scratch "oracle" on that metric. We include all details in Sections A.2 including hyperparameters but we note that, throughout the paper, we tune hyperparameters separately for each budget and localization strategy.

**Findings** From Figure 2 we observe the following. First, when updating (almost) all parameters, **Deepest** achieves strong results on all metrics, which is expected since, at that end of the spectrum, Deepest paired with Reset + Finetune amounts to retraining (almost) the entire model. However, for smaller budgets, it performs very poorly in unlearning metrics as resetting only deeper layers does not cause a sufficient accuracy drop on the forget set (though it at least preserves test accuracy). On the other hand, **Shallowest** is much more effective at unlearning compared to Deepest, and in fact a strong baseline, suggesting that research should consider this baseline alongside EU-k of Goel et al. (2022). This is perhaps due to resetting earlier layers causing larger "disruption" to the information flow in the network. Indeed, contrary to Deepest, this strategy leads to poor test accuracy for several budgets, which is the main downside of this approach. **CritMem** is unable to reach good unlearning quality in this setup. Note that we can only evaluate CritMem for small budgets, as the algorithm of Maini et al. (2023) terminates once enough parameters are reset such that the prediction is flipped to an incorrect one (so the max value of $\alpha$ we can consider is capped). We find that resetting only those parameters is insufficient for achieving good unlearning results when paired with this unlearning algorithm. **SalLoc** has a similar trend to Shallowest in terms of unlearning metrics but leads to higher test accuracy. It performs similarly to CritMem in the range where we can compare them.

Overall, we observe that data-agnostic localization strategies yield poor performance: each of Deepest and Shallowest can either achieve good forget accuracy / MIA efficacy at the expense of utility, or the other way around, but not both. We hypothesize that a data-dependent approach with a finer-grained control of *which* earlier and deeper parameters to update in a way that is informed by the forget set, would yield better trade-offs, due to causing minimal and targeted "disruption" that preserves utility. We have considered two such data-dependent strategies so far, CritMem and SalLoc. CritMem, directly adapted from the memorization literature, results in updating only a small number of parameters (based on its termination criterion) and is unable to achieve good forgetting quality within that budget in the context of localized unlearning, while it is also computationally expensive, as discussed above, due to its iterative nature. SalLoc, on the other hand, is more efficient and improves a little over Shallowest in terms of causing less of a test accuracy drop.

## 5 IMPROVED LOCALIZED UNLEARNING

In this section, building on our previous observations, we take a deeper dive into key design choices of CritMem and SalLoc, formulate hypotheses about their appropriateness and empirically investigate the axes in which they differ, leading to an improved localized unlearning approach.

## 5.1 EXAMINING KEY BUILDING BLOCKS OF LOCALIZATION ALGORITHMS

As discussed in Section 4, SalLoc has the finest granularity, making a decision for whether or not to select individual parameters, whereas CritMem operates on the level of output channels, thus deciding whether the mask should include or exclude all parameters associated with a channel, as a unit. We hypothesize that, because attempts to locate where memorization occurs are based on heuristics and are imperfect, the easier task of making coarser-grained decisions, as in CritMem, is a more appropriate choice compared to the harder, and thus more error-prone, task of making parameter-level assessments of criticality. Specifically, marking all parameters of a channel as critical, if a subset of them are deemed to be critical, can be seen as useful "smoothing", at the potential expense of leading to resetting some more parameters than may have been strictly necessary.

Further, we also hypothesize that CritMem's criticality criterion that uses weighted gradients rather than simply considering the magnitude of gradients themselves (SalLoc) is also more appropriate: it can be seen as a more conservative choice, which is more suitable in light of the heuristic nature of determining criticality. Intuitively, a weight with a small value can be seen as less critical "overall" (for the training set) which is an important signal to consider in addition the gradients on specific forget set examples, as it may also serve as a useful "regularizer" when making heuristic assessments.

Based on the above, we hypothesize that CritMem's granularity level and its criticality criterion are more suitable than SalLoc's for localized unlearning. However, CritMem's iterative nature makes it computationally expensive. Recall that, for each given example, it determines the next most critical channel *one at a time*, resets it and repeats to find the next most critical one, until the label of the example flips. To instead obtain a more efficient localization strategy, we investigate incorporating CritMem's granularity and its criticality criterion into an algorithm that, akin to SalLoc, estimates the critical parameters for each batch in $\mathcal{S}$ in "one-shot", leading to increased efficiency both due to batching over examples in $\mathcal{S}$ as well as due to determining criticality non-iteratively for each batch.

Table 1: Combining different granularity and criticality criteria in a non-iterative localization algorithm. The top-left cell corresponds to the SalLoc, while the bottom-right cell (shaded) represent our proposed approach.

|  |  | Grads | Weighted grads |
|---|---|---|---|
| Parameter | $\Delta_{forget}$ | $-7.58_{\pm 8e-3}$ | $15.07_{\pm 0.01}$ |
|  | $\Delta_{test}$ | $3.63_{\pm 4e-3}$ | $11.55_{\pm 6e-3}$ |
|  | $\Delta_{MIA}$ | $8.74_{\pm 1.10}$ | $-16.32_{\pm 1.39}$ |
| Channel | $\Delta_{forget}$ | $-12.33_{\pm 8e-3}$ | $1.58_{\pm 0.01}$ |
|  | $\Delta_{test}$ | $1.55_{\pm 2e-3}$ | $4.41_{\pm 2e-3}$ |
|  | $\Delta_{MIA}$ | $15.53_{\pm 1.10}$ | $3.33_{\pm 1.28}$ |

In Table 1, we investigate the effect of the granularity and criticality criteria mentioned above, in the context of batched and non-iterative localization algorithms, on the same experimental setup (dataset, forget set, etc) as in Section 4. We find that indeed the best choice is given by using the channel-wise granularity and weighted gradients. We therefore build on these decisions to devise our localization strategy in the next section.

## 5.2 INTRODUCING OUR LOCALIZATION STRATEGY

Given a forget set $\mathcal{S}$ and an original model $\theta^o$ with $p$ parameters, for $j \in \{1, \ldots, p\}$, let $\theta^o_j$ and $g_j(\theta^o, \mathcal{S})$ represent the weight and gradient values on the forget set, respectively. We define the *criticality score* $s_j$ of the $j^{th}$ parameter as the magnitude of the weighted gradient over the forget set: $s_j = |\theta^o_j \cdot g_j(\theta^o, \mathcal{S})|$; this is the same criticality criterion used in CritMem, whereas SalLoc simply considers the magnitude of the gradient for each parameter $|g_j(\theta^o, \mathcal{S})|$ for assessing its criticality.

As discussed above, we choose to determine criticality in a coarser-grained way compared to individual parameters. To that end, for an output "channel" $o_i$ (or "neuron" more broadly, encompassing non-convolutional architectures), we describe how to obtain its criticality score $c_{o_i}$ based on the criticality score of its constituent parameters. Let $\tilde{s}_i$ be a list of the criticality scores for the parameters belonging to neuron $o_i$, sorted in descending order. We set the neuron criticality $c_{o_i}$ to be the average of the top $h$ scores of its associated parameters: $c_{o_i} = \frac{1}{h}\Sigma_{j=1}^h \tilde{s}_i[j]$.

Finally, having obtained the neuron criticality scores, we put together the mask $m_\alpha$ for parameter budget $\alpha$, represented as a binary vector of size $p$, where a 1 indicates the corresponding parameter will be updated by the unlearning algorithm, whereas an entry of 0 indicates it will be kept unchanged. To this end, we form another sorted list $\tilde{c}$, that sorts the neurons in descending order of their criticality scores. We then pick the largest number of neurons from the start of the sorted list, such that the total

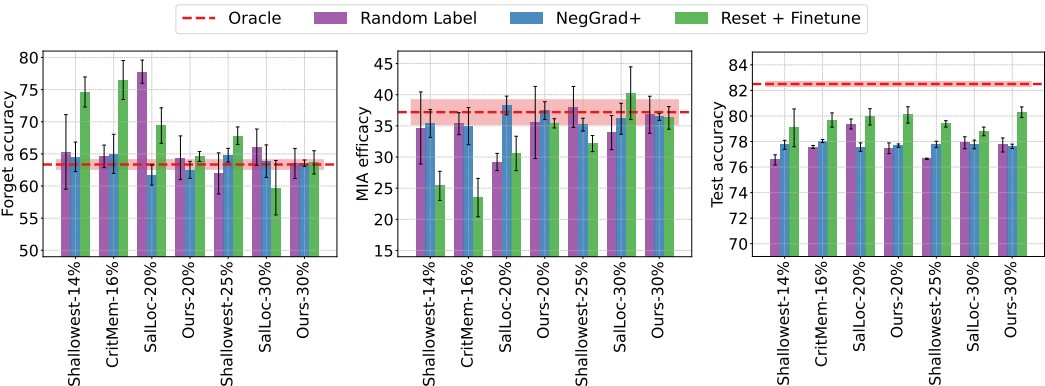

Figure 3: Pairing localization strategies / budgets (e.g. Ours-30% denotes applying our localization strategy to select 30% of parameters) with three unlearning algorithms, on CIFAR-10 / ResNet (the ideal behaviour is to match the "Oracle"). **Our method has the best unlearning efficacy, paired with *any* unlearning algorithm, and its performance degrades much less than SalLoc's when the budget reduces from 30% to 20%; meanwhile it has no worse (or better) test accuracy**.

selected parameters are within the budget. Then, we assign a $1$ to all entries of $m_\alpha$ for parameters belonging to the chosen critical neurons and $0$ to the rest. We provide pseudocode in Section A.5.

Our localization strategy can, in principle, be paired with any unlearning algorithm, but we find we obtain strongest results by pairing it with the simple Reset + Finetune (RFT) algorithm. We refer to the combination of our localization strategy with RFT as **Deletion by Example Localization (DEL)**.

## 6 Comparison to state-of-the-art and analyses

In this section, we carry out comprehensive experiments on two datasets and architectures (CIFAR-10 with ResNet-18 and SVHN with ViT; see Section A.2 for details), to examine the performance of our localization strategy paired with various unlearning algorithms, on different types of forget sets, as well as various analyses to understand the factors behind the success of localized unlearning methods.

**Pairing with different unlearning algorithms.** In Figure 3, we compare different localization strategies, for different values of the parameter budget, paired with three different unlearning algorithms. We observe that i) our method pairs well with different unlearning algorithms, ii) in terms of forget accuracy and MIA efficacy, our method yields the best results, coming very close to the ideal "oracle" reference point by updating only 30% of the network's parameters, iii) at the same time, test accuracy is no worse (and sometimes better) using our method.

**DEL is robust to the parameter budget.** Figure 2 shows that DEL is more robust to the budget compared to all other strategies considered, yielding strong results across several budgets. Figure 3 corroborates this finding in the context of different unlearning algorithms too, showing that our localization method experiences much lower performance degradation compared to SalLoc, when the budget is reduced: we significantly outperform SalLoc when the budget is 20%.

**DEL outperforms the state-of-the-art for both datasets and forget set types.** We compare DEL to state-of-the-art methods for unlearning, including ones that update all parameters on two different datasets. On CIFAR-10, we also consider two forget sets: an **IID forget set** comprised of 10% of randomly-chosen training samples, and a **non-IID forget set** of the same size but choosing samples belonging to a subset of the classes (see details in Section A.2). We present the results in Table 2 and Figure 4 (and Table 8 in the Appendix). For each localized unlearning strategy and on each dataset, we report results using its best identified parameter budget and its best-paired unlearning algorithm for that setting. We observe that, generally, localized unlearning methods outperform their full-parameter counterparts in terms of unlearning metrics ($\Delta_{forget}$ and $\Delta_{MIA}$); on the former ($\Delta_{forget}$) because full-parameter methods end up causing the forget set accuracy of unlearning to become significantly lower than the ideal reference point, yielding a negative $\Delta$ (associated with poor MIA efficacy, too). We hypothesize that localized unlearning, due to making a more targeted update, can be more easily tuned to reach the desired reference point for the forget accuracy rather

Table 2: DEL outperforms the state-of-the-art localized and full-parameter unlearning, on CIFAR-10/ResNet-18 on two forget sets: the **Non-IID** forget set consists of $10\%$ of the training samples, randomly selected from two classes, whereas the **IID** one consists of $10\%$ of the training samples, randomly selected from all classes. We use three metrics, each represented as $\Delta$, obtained by subtracting the unlearning algorithm's value for a given metric from the Oracle's value for that metric): forget accuracy ($\Delta_{\text{forget}}$), MIA efficacy ($\Delta_{\text{MIA}}$) and test accuracy ($\Delta_{\text{test}}$). Note that SalLoc-RL corresponds to "SalUn" which employs Random Label, regarded as state-of-the-art.

| | | **Non-IID Forget Set** | | | **IID Forget Set** | | |
|---|---|---|---|---|---|---|---|
| | | $\Delta_{\text{forget}}$ | $\Delta_{\text{MIA}}$ | $\Delta_{\text{test}}$ | $\Delta_{\text{forget}}$ | $\Delta_{\text{MIA}}$ | $\Delta_{\text{test}}$ |
| **Full-parameter Unlearning** | Retraining(Oracle) | $0.00_{\pm0.00}$ | $0.00_{\pm0.00}$ | $0.00_{\pm0.00}$ | $0.00_{\pm0.00}$ | $0.00_{\pm0.00}$ | $0.00_{\pm0.00}$ |
| | Fine-tuning | $-5.60_{\pm0.89}$ | $6.07_{\pm1.10}$ | $-1.61_{\pm0.34}$ | $-1.98_{\pm1.10}$ | $1.96_{\pm1.11}$ | $-2.00_{\pm0.72}$ |
| | NegGrad+ | $-4.44_{\pm0.95}$ | $4.92_{\pm1.14}$ | $4.61_{\pm0.22}$ | $1.87_{\pm1.31}$ | $1.89_{\pm1.30}$ | $4.21_{\pm0.67}$ |
| | NegGrad | $-3.30_{\pm0.72}$ | $3.76_{\pm0.94}$ | $4.60_{\pm0.18}$ | $-15.04_{\pm0.40}$ | $15.39_{\pm0.39}$ | $-1.11_{\pm0.51}$ |
| | Random Label | $-1.64_{\pm0.98}$ | $2.07_{\pm1.15}$ | $4.33_{\pm0.19}$ | $1.69_{\pm0.46}$ | $1.69_{\pm0.47}$ | $4.93_{\pm0.47}$ |
| | L1-sparse | $-1.50_{\pm0.82}$ | $-1.01_{\pm1.03}$ | $2.07_{\pm0.51}$ | $1.80_{\pm1.20}$ | $-1.80_{\pm1.08}$ | $0.62_{\pm0.67}$ |
| | IU | $-5.00_{\pm0.88}$ | $5.04_{\pm0.91}$ | $4.18_{\pm0.19}$ | $-2.20_{\pm0.39}$ | $2.19_{\pm0.38}$ | $10.94_{\pm0.43}$ |
| **Localized Unlearning** | SSD | $-11.16_{\pm6.28}$ | $11.18_{\pm6.29}$ | $2.68_{\pm1.18}$ | $1.60_{\pm1.99}$ | $1.59_{\pm1.98}$ | $11.58_{\pm1.03}$ |
| | CritMem-RL ($\alpha = 16\%$) | $-1.82_{\pm1.19}$ | $1.87_{\pm1.21}$ | $4.86_{\pm0.19}$ | $-2.03_{\pm0.45}$ | $2.05_{\pm0.45}$ | $4.36_{\pm0.37}$ |
| | Shallowest-RL ($\alpha = 25\%$) | $1.29_{\pm1.62}$ | $-0.80_{\pm1.74}$ | $5.88_{\pm0.18}$ | $3.41_{\pm0.77}$ | $-3.43_{\pm0.78}$ | $6.43_{\pm0.52}$ |
| | SalLoc-RL ($\alpha = 30\%$) | $-2.8_{\pm1.45}$ | $3.30_{\pm1.54}$ | $4.63_{\pm0.27}$ | $-3.81_{\pm0.40}$ | $3.80_{\pm0.39}$ | $4.29_{\pm0.45}$ |
| | **DEL ($\alpha = 30\%$)** | $0.43_{\pm1.06}$ | $0.64_{\pm1.23}$ | $2.23_{\pm0.25}$ | $0.97_{\pm0.42}$ | $-0.97_{\pm0.40}$ | $1.87_{\pm0.49}$ |

Table 3: **Random-vs-standard masking** using RFT unlearning, $\alpha$=16%, on CIFAR-10 / ResNet-18.

| | **Standard Masking** | | | **Random Masking** | | |
|---|---|---|---|---|---|---|
| | CritMem | SalLoc | Ours | CritMem | SalLoc | Ours |
| $\Delta_{\text{forget}}$ | $-13.25_{\pm1.53}$ | $-8.26_{\pm0.92}$ | $-2.86_{\pm1.09}$ | $-18.60_{\pm1.03}$ | $-13.26_{\pm0.80}$ | $-8.58_{\pm0.98}$ |
| $\Delta_{\text{MIA}}$ | $13.73_{\pm1.62}$ | $8.75_{\pm1.09}$ | $3.36_{\pm1.27}$ | $19.09_{\pm1.21}$ | $13.78_{\pm1.02}$ | $9.07_{\pm1.17}$ |
| $\Delta_{\text{test}}$ | $2.85_{\pm0.31}$ | $3.50_{\pm0.53}$ | $2.62_{\pm0.22}$ | $1.99_{\pm0.19}$ | $1.91_{\pm0.57}$ | $1.89_{\pm0.21}$ |

than "overshooting" it, amending the above issue. On the other hand, full-parameter methods lead to (marginally) better test accuracy in some cases, especially for the IID forget set. Out of the considered localized methods, that DEL outperforms the state-of-the-art on all metrics and across forget sets.

**Localized unlearning succeeds due to selecting critical parameters.** We design a control experiment to investigate to what extent the success of localized unlearning is dependent on *which* parameters are chosen (rather than simply *how many*). To this end, we compare the mask produced by each localization strategy to a "random mask" that is constructed to follow the same structure and distribution of the number of chosen parameters per layer as the corresponding non-random mask. For example, to create the random mask that CritMem will be compared with, we randomly select a number of channels for each layer equal (but randomly selected this time) to the number of channels that CritMem selects for that layer. The results in Table 3 indicate that, beyond doubt, localized unlearning algorithms succeed due to pinpointing critical parameters.

**Is localized unlearning better due to tailoring to $\mathcal{S}$?** We design a set of experiments to investigate this by changing the criticality criterion. Specifically, we choose two criteria that are not specific to the forget set $\mathcal{S}$: the first uses only the magnitude of the weights ("weights"), and the second uses weighted gradients, but where the gradients this time are over all of $\mathcal{D}_{\text{train}}$ rather than just $\mathcal{S}$ ("Weighted gradients (train set)"). Our rationale is that, if either of these forget-set-agnostic criteria work equally well as our method's criterion ("Weighted gradients (forget set)"), this would suggest that the success of our method is not due to specialization to $\mathcal{S}$ but rather finding parameters that are "generally critical" for the training data. We observe from Table 4 that, for the IID forget set, the above two criteria that depend on $\mathcal{D}_{\text{train}}$ rather than $\mathcal{S}$ specifically, yield more similar results to our criterion. This is reasonable since the forget and train follow the same distribution in the IID forget set case. On the other hand, for the non-IID forget set, we do observe that tailoring the criticality criterion used in the localization strategy to $\mathcal{S}$ yields better results in terms of the unlearning metrics. These findings suggest that i) the success of different localization strategies is dependent on the distribution of the forget set, ii) our method (shaded gray area in the table) is a top performer in all cases.

Table 4: Investigation of different granularity and criticality criteria in a non-iterative localization algorithm ($\alpha$=15%) on **IID** and **Non-IID forget set**. The shaded region corresponds to our method.

| Granularity | | Non-IID forget set | | | | IID forget set | | | |
|---|---|---|---|---|---|---|---|---|---|
| | | Gradients (forget set) | Weights | Weighted gradients (train set) | Weighted gradients (forget set) | Gradients (forget set) | Weights | Weighted gradients (train set) | Weighted gradients (forget set) |
| Individual parameter | $\Delta_{\text{forget}}$ | $-7.58_{\pm 8e-3}$ | $15.50_{\pm 6e-3}$ | $14.03_{\pm 0.01}$ | $15.07_{\pm 0.01}$ | $-6.72_{\pm 0.41}$ | $11.67_{\pm 0.51}$ | $11.19_{\pm 0.15}$ | $11.15_{\pm 0.48}$ |
| | $\Delta_{\text{test}}$ | $3.63_{\pm 4e-3}$ | $10.80_{\pm 5e-3}$ | $10.60_{\pm 6e-3}$ | $11.55_{\pm 6e-3}$ | $1.98_{\pm 0.45}$ | $11.11_{\pm 0.60}$ | $10.90_{\pm 0.51}$ | $11.05_{\pm 0.54}$ |
| | $\Delta_{\text{MIA}}$ | $8.74_{\pm 1.10}$ | $-14.73_{\pm 0.97}$ | $-13.89_{\pm 1.29}$ | $-16.32_{\pm 1.39}$ | $6.81_{\pm 0.41}$ | $11.68_{\pm 0.50}$ | $-11.21_{\pm 0.52}$ | $-11.13_{\pm 0.46}$ |
| Output channel | $\Delta_{\text{forget}}$ | $-12.33_{\pm 8e-3}$ | $6.02_{\pm 5e-3}$ | $-2.52_{\pm 0.01}$ | $1.58_{\pm 0.01}$ | $-5.18_{\pm 0.40}$ | $1.33_{\pm 0.41}$ | $1.69_{\pm 0.68}$ | $0.65_{\pm 0.60}$ |
| | $\Delta_{\text{test}}$ | $1.55_{\pm 2e-3}$ | $5.64_{\pm 2e-3}$ | $3.52_{\pm 4e-3}$ | $4.41_{\pm 2e-3}$ | $1.26_{\pm 0.54}$ | $2.86_{\pm 0.45}$ | $2.27_{\pm 0.44}$ | $3.05_{\pm 0.43}$ |
| | $\Delta_{\text{MIA}}$ | $15.53_{\pm 1.10}$ | $-1.02_{\pm 0.93}$ | $6.73_{\pm 1.12}$ | $3.33_{\pm 1.28}$ | $5.16_{\pm 0.39}$ | $-1.40_{\pm 0.39}$ | $1.67_{\pm 0.68}$ | $0.69_{\pm 0.59}$ |

# 7 DISCUSSION AND CONCLUSION

**To summarize**, we have performed an investigation on whether hypotheses for where memorization happens in the network give rise to improved localized unlearning. Our investigation led us to propose a new localization strategy that is more practical and efficient compared to the algorithm from the memorization literature that it builds upon, while outperforming the state-of-the-art unlearning methods on several metrics, when paired with different unlearning algorithms. Our proposed DEL method, obtained by pairing our strategy with the simple RFT unlearning algorithm, sets a new state-of-the-art on forget sets of different distributions, different datasets and architectures, and across parameter budgets. We find that for non-IID forget sets, tailoring the parameter selection to the specific forget set (rather than the training set more broadly) is more important than it is for IID forget sets. Our method outperforms others in both cases but to different degrees, pointing to important questions for future work regarding what other characteristics of forget sets affect the behaviours and success rates of localized unlearning.

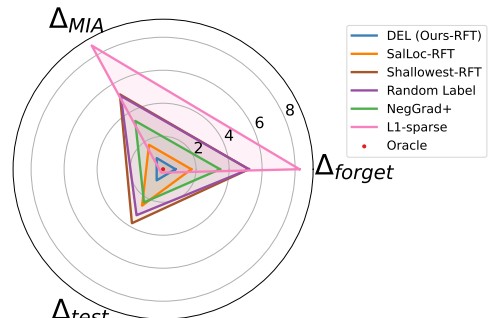

Figure 4: **On SVHN with ViT**, **DEL outperforms state-of-the-art full-parameter and localized unlearning in terms of unlearning quality**. L1-sparse has better test accuracy than DEL but has poor unlearning performance. These results are for the non-IID forget set, and $\alpha = 30\%$ for localized methods; see Table 8 for full results.

**So, does memorization inform unlearning?** Hase et al. (2024) find that, for model editing in LLMs, the "causal tracing" method (Meng et al., 2022) for knowledge localization, surprisingly, does not indicate which layer to modify in order to most successfully rewrite a stored fact with a new one. That is, they find that success in editing tasks is generally unrelated to localization results based on causal tracing. Guo et al. study whether mechanistic interpretability insights improve unlearning of "factual associations" in LLMs. They also find that localization techniques based on preserving outputs (such as causal tracing) yield performance that is no better, or even worse, than non-localized unlearning. However, they come up with a mechanistic unlearning method that does outperform both output-based localization and non-localized unlearning, showing that some form of localization is useful. Our results, in the very different context of unlearning a subset of data in vision classifiers, offer an important data point in this ongoing discussion. In line with Hase et al. (2024), we find that directly translating memorization hypotheses into localization strategies does not help unlearning: Deepest led to very poor unlearning results (demonstrating either its weakness as a memorization locator, or the disconnect between memorization localization and unlearning performance), and CritMem, while showing more promise, did not perform better than simple baselines, while being significantly more expensive than them. However, insights from (Maini et al., 2023), in particular regarding the granularity and criticality criterion used during localization led us to improve upon the state-of-the-art localized and full-parameter unlearning methods, renewing hopes that, while memorization localization and unlearning may be separate research questions, progress in the former may guide progress in the latter.

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

# A  APPENDIX

## A.1  UNLEARNING DEFINITION

In this section, we discuss an alternative formal definition of unlearning, proposed in in Ginart et al. (2019); Neel et al. (2021), using a notion closely related to Differential Privacy (Dwork, 2006).

**Definition A.1. Unlearning-2.** For a training algorithm $\mathcal{A}$, an algorithm $\mathcal{U}$ is an $(\epsilon, \delta)$-*unlearner* if, for any training dataset $\mathcal{D}_{\text{train}}$ and forget set $\mathcal{S}$, the distributions of $\mathcal{A}(\mathcal{D}_{\text{train}} \setminus \mathcal{S})$ and $\mathcal{U}(\theta^o, \mathcal{S}, \mathcal{D}_{\text{train}} \setminus \mathcal{S})$ are $(\epsilon, \delta)$-close, where we say two distributions $\mu, \nu$ are $(\epsilon, \delta)$-*close* if $\mu(B) \leq e^\epsilon \nu(B) + \delta$ and $\nu(B) \leq e^\epsilon \mu(B) + \delta$ for all measurable events $B$.

Intuitively, the above compares (the distribution of) models that are obtained by two different recipes to one another:

- $\mathcal{A}(\mathcal{D}_{\text{train}} \setminus \mathcal{S})$, retraining "from scratch" on only the retain set, which is prohibitively expensive but ideal from the standpoint of eliminating the influence of $\mathcal{S}$ on the model, and

- $\mathcal{U}(\theta^o, \mathcal{S}, \mathcal{D}_{\text{train}} \setminus \mathcal{S})$, applying $\mathcal{U}$ to post-process the original model $\theta^o$ in order to unlearn $\mathcal{S}$.

We desire unlearning algorithms $\mathcal{U}$ that cause these two recipes to yield similar models, with the second recipe being substantially more computationally-efficient compared to the first, in order to justify paying the cost of approximate unlearning rather than simply using the first recipe directly.

Note that we refer to *distributions* here since re-running either of the two recipes with a different random seed, that controls the initialization or the order of mini-batches, for example, would yield slightly different model weights in each case. The above definition therefore measures unlearning quality based on the notion of $(\epsilon, \delta)$-*closeness* between the two distributions. The smaller $\epsilon$ and $\delta$ are, indicating increased closeness, the better the unlearning algorithm.

**Relationship and differences to our definition** This definition compares distributions in weight space, whereas our Definition 2.1 compares distributions of *outputs* of models on the forget set. We opted for the latter in the main paper as it more closely reflects the metrics we use for evaluation (which are the standard metrics used in unlearning papers). Note that, even works that adopt definitions in weight-space end up operationalizing them using outputs of models (Triantafillou et al., 2024) instead of performing weight-space comparisons. This is for several reasons: comparing weights of models directly may be inappropriate since neural networks are permutation-invariant. Weight space is also much higher dimensional, posing challenges in creating the right metrics, and, finally, ultimately what we may care about for various applications of interest is the "behaviours" (e.g. predictions) of models, rather than their weights. Definition 2.1 captures this more directly.

## A.2  EXPERIMENTAL SETUP

**Datasets** The CIFAR-10 dataset (Krizhevsky et al., 2009) consists of $50,000$ train and $10,000$ test images of shape $32 \times 32$ from 10 classes. The SVHN dataset (Netzer et al., 2011) includes $73,257$ train and $26,032$ test samples. The samples are of shape $32 \times 32$ pixel, and from 10 classes. The ImageNet-100 (Hugging Face version) dataset is a subset of ImageNet (Deng et al., 2009), containing $126,689$ train and $5,000$ test samples from 100 classes, randomly selected from the original ImageNet classes. The resolution of the images on the shortest side is 160 pixels.

We perform no preprocessing or augmentation on the images of CIFAR-10 and SVHN, except dividing the feature values by 255. For ImageNet-100, on the other hand, we randomly crop the train images to size $128 \times 128$, and horizontally flip them. Moreover, we first resize the test images to $160 \times 160$, and then center-crop them to $128 \times 128$. We normalize the features of both train and test images with the mean and variance of ImageNet.

For the unlearning experiments, we employ two different forget sets: (1) IID, where we uniformly select approximately 10 percent of the images from the train set, and (2) NonIID, in which we randomly select half of the samples from two classes (2 and 5 in CIFAR-10, and 3 and 6 in SVHN) so that the size of the forget set is almost 10 percent of the train set. Note that in the former, the distributions of the samples in the forget and retain sets are highly similar, whereas in the latter, the sample distribution of the forget set is very different from that of the retain set.

**Models**   We capitalize on the original implementation of ResNet-18 and ResNet-50 (He et al., 2016) from PyTorch and the implementation of Vision Transformer (ViT) (Dosovitskiy, 2020) from (Wang, 2021). ResNet-18 and ViT contain around 11 million parameters, whereas ResNet-50 has approximately 25 million parameters. Due to the low-resolution nature of CIFAR-10, we replace the first convolutional layer of ResNet-18 with a new convolutional layer with kernel size of $3\times3$, and remove the max-pooling layer.

Note that the architectures of the considered models are very different from each other. ResNet-18/50 are convolutional (Conv) networks with an input Conv layer, multiple residual blocks, and a final classifier layer. The normalization layer of ResNet-18/50 is batch normalization (Ioffe & Szegedy, 2015). In ResNet-18/50, the input images are downsampled multiple times so that deeper layers operate on smaller input tensors. However, deeper layers have more filters (and thus, more trainable parameters) than shallower layers.

The ViT architecture, on the other hand, employs linear (fully-connected) and multi-head attention layers as its main building blocks. It first divides the input images into square patches (e.g. of shape $8\times8$) and give them as tokens (after some preprocessing including positional encoding) to the encoder blocks. No downsampling is performed on the input tensors by the encoder blocks. Moreover, all (i.e. both deeper and shallower) encoder blocks have identical number of trainable parameters. The normalization layer of ViT architectures is layer normalization (Ba et al., 2016).

**Training**   For the original (pretrained) models, we train ResNet-18 on CIFAR-10 and ViT on SVHN (i.e. on the training set of the datasets) for 50 epochs using the SGD optimizer with momentum of 0.9, cross-entropy loss function, and batch size of 128. The base learning rate values are 0.1 and 0.05 for ResNet-18-CIFAR-10 and ViT-SVHN, respectively, which is gradually decayed by factor of 0.01 using the Cosine Annealing scheduler. For the oracle model (gold standard), we train the model from scratch only on the retain set, following the same procedure employed for the pretrained model, except the number of epochs, which we set to 20, and learning rate, which is the half of that in original training. We provide the hyper-parameter values for the approximate unlearning algorithms in the tables below. We repeat each experiment three times and report the average values along with $95\%$ confidence interval margins of the mean.

Table 5: Learning rate tuning.

|  | Scheduler | Parameters |
|---|---|---|
| Finetuning/ l1-sparse | CosineAnnealingLR | $\eta_{min} = 0.01 * lr_{init}$ |
| Random Label | CosineAnnealingLR | $\eta_{min} = 0.5 * lr_{init}$ |
| NegGrad+/NegGrad | Constant | - |

|  | Non-IID forget set | | IID Forget set | |
|---|---|---|---|---|
|  | $lr_{best}$ | candidate values | $lr_{best}$ | Candidate values |
| Finetuning | 1.25 | [0.5, 1.5] | 1.25 | [0.5, 1.5] |
| l1-sparse | 0.5 | [0.1, 1] | 0.5 | [0.1, 1] |
| Random Label | 7e-3 | [5e-3, 1e-2] | 6e-3 | [5e-3, 1e-2] |
| NegGrad+ | 7e-4 | [5e-4, 1e-3] | 0.14 | [0.1, 1] |
| NegGrad | 7e-6 | [5e-6, 1e-5] | 4e-3 | [1e-3, 5e-3] |
| CritMem-RL ($\alpha = 16\%$) | 0.02 | [0.01, 0.1] | 0.02 | [0.01, 0.1] |
| Shallowest-RL ($\alpha = 25\%$) | 7e-3 | [5e-3, 1e-2] | 7e-3 | [5e-3, 1e-2] |
| SalLoc-RL ($\alpha = 30\%$) | 0.012 | [5e-3, 1e-2] | 0.012 | [5e-3, 1e-2] |
| DEL ($\alpha = 30\%$) | 0.015 | [5e-3, 1e-2] | 0.015 | [5e-3, 1e-2] |

## A.3   METRICS

Following (Fan et al., 2023), we employ accuracy and membership inference attack (MIA) efficacy to evaluate the effectiveness of different unlearning algorithms.

To compute MIA efficacy, a support vector classifier (SVC) is trained on top of outputs coming from the unlearned model for the task of predicting whether an example was used in training or not. This is performed through supervised learning where the test set is used as "unseen data" and a

subset of retain set (with the same size as and similar label distribution to the test set) as "seen data". Specifically, the SVC is trained on the "prediction" outputs (i.e. the integers representing the index of the predicted class), aiming to distinguish predictions from seen versus unseen data. Then, the trained classifier is utilized to predict if each sample in the forget set belongs to the seen or unseen data on the unlearned model. Given that, MIA efficacy is computed as follows:

$$MIA_{\text{efficacy}} = \frac{TN}{|\mathcal{S}|},$$

where TN is the number of true negatives, i.e. forget samples that the classifier recognizes as likely unseen data for the unlearned model, and $|\mathcal{S}|$ is the size of the forget set.

Intuitively, a higher value of $MIA_{\text{efficacy}}$ means that the unlearned model has been more successful in "fooling" the SVC classifier (i.e. the "membership inference attacker") into thinking that the forget set was not used in training. However, to interpret how high we expect $MIA_{\text{efficacy}}$ to be for an unlearned model, we must consult the reference point of how high this quantity would be for a model retrained from scratch without the forget set. Note that, even in that case of "ideal unlearning", $MIA_{\text{efficacy}}$ is not necessarily 100%, and in fact it can be much lower than this. This is because, some examples in the forget set might be so "easy" that, even without ever seeing them, the retrained model can still be equally accurate on those examples as it would have been if they were actually included in training. This would lead to its $MIA_{\text{efficacy}}$ being lower, since some forget set examples would be classified as "seen" by the SVC. For this reason, in our experiments, we use the reference point as the $MIA_{\text{efficacy}}$ obtained from retrain-from-scratch as the optimal value for this metric. An ideal unlearning algorithm would therefore match that value.

## A.4 MIA Evaluation

In this section, we present MIA evaluation results using various MIAs for each model-dataset combination. We compare two MIAs that leverage the model's (i) correctness (Table 7) and (ii) confidence (Table 6). Specifically, we train an SVC to distinguish between the seen (train) and unseen (test) data using either (i) the predictions (i.e. the integers representing the index of the predicted class) of the unlearned model on retain and test examples or (ii) the confidences (i.e. the Softmax values associated with these predicted labels) of the unlearned model on these examples. The results for the first variant (correctness-based MIA) are already presented in Tables 2 and 8, with a summary provided in Table 7 in this section. Here, we expand the MIA evaluations by incorporating the second variant (confidence-based MIA).

According to Table 6 and 7, the absolute value of $\Delta_{MIA}$ values are larger when using the confidence-based MIA compared to the correctness-based MIA. Since the model's confidence on retain and test samples provides more information than its predictions on these samples, providing the SVC with confidence values results in a stronger MIA than using the predictions.

Additionally, in the ResNet-18–CIFAR-10 setting, we observe that our localized unlearning algorithm significantly outperforms the other comparison methods across all evaluated MIAs for both IID and non-IID forget sets. Similarly, in the ViT-SVHN setting with non-IID forget sets, our method outperforms the other methods in both correctness-based and confidence-based MIA evaluations. However, when the forget set is IID in ViT-SVHN setting, for both MIA variants there exists a full-parameter unlearning algorithm (sometimes multiple, depending on the type of MIA) that outperforms all the localized unlearning algorithms, including our method, in terms of MIA evaluation. For example, Fine-tuning yields a more effective confidence-based MIA, while Fine-tuning, Random Label, and L1-sparse demonstrate enhanced performance in correctness-based MIA compared to the localized unlearning algorithms. This is an interesting observation that we hope future work will investigate further.

Table 6: **Confidence**-based MIA evaluation ($\Delta_{MIA}$)

| | | ResNet-18 - CIFAR-10 | | ViT - SVHN | |
| --- | --- | --- | --- | --- | --- |
| | | IID | Non-IID | IID | Non-IID |
| **Full-parameter Unlearning** | Retraining(Oracle) | $0.00_{\pm 0.00}$ | $0.00_{\pm 0.00}$ | $0.00_{\pm 0.00}$ | $0.00_{\pm 0.00}$ |
| | Fine-tuning | $6.24_{\pm 2.18}$ | $6.23_{\pm 1.03}$ | $-1.97_{\pm 0.32}$ | $13.85_{\pm 0.90}$ |
| | NegGrad+ | $3.69_{\pm 1.34}$ | $11.27_{\pm 1.19}$ | $-19.40_{\pm 11.60}$ | $3.75_{\pm 1.82}$ |
| | NegGrad | $28.18_{\pm 0.87}$ | $10.44_{\pm 0.83}$ | $12.47_{\pm 0.25}$ | $29.02_{\pm 1.03}$ |
| | Random Label | $-31.55_{\pm 1.30}$ | $-19.96_{\pm 1.46}$ | $-15.56_{\pm 2.45}$ | $-6.44_{\pm 3.58}$ |
| | L1-sparse | $4.35_{\pm 0.88}$ | $10.49_{\pm 2.17}$ | $-7.74_{\pm 0.58}$ | $7.62_{\pm 0.75}$ |
| **Localized Unlearning** | CritMem-RL ($\alpha = 16\%, 1\%$) | $-31.56_{\pm 1.04}$ | $-14.08_{\pm 2.58}$ | $13.12_{\pm 0.24}$ | $32.80_{\pm 0.71}$ |
| | Shallowest-RL/RFT ($\alpha = 25\%, 30\%$) | $-25.87_{\pm 0.89}$ | $-18.39_{\pm 1.45}$ | $-16.83\pm1.05$ | $-2.5_{\pm 8.54}$ |
| | SalLoc-RL/RFT ($\alpha = 30\%$) | $-24.24_{\pm 0.89}$ | $-14.15_{\pm 1.76}$ | $-17.77_{\pm 0.42}$ | $1.67_{\pm 0.92}$ |
| | **DEL** ($\alpha = 30\%$) | $-0.59_{\pm 0.90}$ | $-0.90\pm1.20$ | $-5.48_{\pm 0.64}$ | $1.12_{\pm 0.76}$ |

Table 7: **Correctness**-based MIA evaluation ($\Delta_{MIA}$)

| | | ResNet-18 - CIFAR-10 | | ViT - SVHN | |
| --- | --- | --- | --- | --- | --- |
| | | IID | Non-IID | IID | Non-IID |
| **Full-parameter Unlearning** | Retraining(Oracle) | $0.00_{\pm 0.00}$ | $0.00_{\pm 0.00}$ | $0.00_{\pm 0.00}$ | $0.00_{\pm 0.00}$ |
| | Fine-tuning | $1.96_{\pm 1.11}$ | $6.07_{\pm 1.10}$ | $-1.05_{\pm 0.42}$ | $11.22_{\pm 0.65}$ |
| | NegGrad+ | $1.89_{\pm 1.38}$ | $4.92_{\pm 1.14}$ | $-6.99_{\pm 3.53}$ | $3.36_{\pm 1.67}$ |
| | NegGrad | $15.39_{\pm 0.39}$ | $3.76_{\pm 0.94}$ | $8.00_{\pm 0.22}$ | $19.70_{\pm 0.53}$ |
| | Random Label | $1.69_{\pm 0.47}$ | $2.07_{\pm 1.15}$ | $-2.82_{\pm 0.46}$ | $5.13_{\pm 1.81}$ |
| | L1-sparse | $-1.80_{\pm 1.08}$ | $-1.01_{\pm 1.03}$ | $-2.43_{\pm 0.39}$ | $8.66_{\pm 0.31}$ |
| **Localized Unlearning** | CritMem-RL ($\alpha = 16\%, 1\%$) | $2.05_{\pm 0.45}$ | $1.87_{\pm 1.21}$ | $8.16_{\pm 0.22}$ | $20.71_{\pm 0.26}$ |
| | Shallowest-RL/RFT ($\alpha = 25\%, 30\%$) | $-3.43_{\pm 0.78}$ | $-0.80_{\pm 1.74}$ | $-6.89_{\pm 0.86}$ | $-5.24_{\pm 2.32}$ |
| | SalLoc-RL/RFT ($\alpha = 30\%$) | $3.80_{\pm 0.39}$ | $3.30_{\pm 1.54}$ | $4.55_{\pm 0.32}$ | $-1.71_{\pm 0.31}$ |
| | **DEL** ($\alpha = 30\%$) | $-0.97_{\pm 0.40}$ | $0.64_{\pm 1.23}$ | $-4.26_{\pm 0.32}$ | $-0.78_{\pm 0.92}$ |

## A.5 PSEUDOCODE

---

**Algorithm 1:** Our localization strategy

**Input:** Original model $\theta^o$ with $p$ parameters, forget set $\mathcal{S}$, parameter budget $\alpha$

**Output:** Localization mask $m_\alpha$

```
/* Compute criticality score for each parameter              */
```
1 $s_j \leftarrow 0, \forall j \in \{1, \ldots, p\}$

2 **for** *Mini-batch* $\mathcal{B} \in \mathcal{S}$ **do**

3 $\quad\lfloor\ s_j = s_j + \theta_j^o \cdot g_j(\theta^o, \mathcal{B}), \forall j \in \{1, \ldots, p\}$

```
/* Consider only the magnitude of each weighted gradient     */
```
4 $s_j \leftarrow |s_j|, \forall j \in \{1, \ldots, p\}$

```
/* Compute criticality score for each output channel/neuron  */
```
5 $\tilde{s} = Sort(s), c_{o_i} = \frac{1}{h}\Sigma_{j=1}^h \tilde{s}_i[j], \forall o_i \in \theta^o$

```
/* Construct localization mask                               */
```
6 $\tilde{c} = Sort(c), m_\alpha = \mathbf{1}\left(\sum_{i=1}^j |\tilde{c}_{o_i}| \leq \left(p \cdot \alpha\right)\right)$

7 **return** Localization mask $m_\alpha$

---

## A.6 ViT-SVHN EXPERIMENTS

The full results corresponding to Figure 4 are detailed in Table 8.

Table 8: Comparison to state-of-the-art, including algorithms that update all parameters ("**Full-Parameter**") on **Non-IID** and **IID** forget set when training a **ViT** model on **SVHN** dataset.

| | | Non-IID Forget Set | | | IID Forget Set | | |
|---|---|---|---|---|---|---|---|
| | | $\Delta_{forget}$ | $\Delta_{MIA}$ | $\Delta_{test}$ | $\Delta_{forget}$ | $\Delta_{MIA}$ | $\Delta_{test}$ |
| **Full-parameter Unlearning** | Retraining(Oracle) | $0.00_{\pm0.00}$ | $0.00_{\pm0.00}$ | $0.00_{\pm0.00}$ | $0.00_{\pm0.00}$ | $0.00_{\pm0.00}$ | $0.00_{\pm0.00}$ |
| | Fine-tuning | $-11.27_{\pm0.66}$ | $11.22_{\pm0.65}$ | $-0.76_{\pm0.20}$ | $-2.73_{\pm0.49}$ | $-1.05_{\pm0.42}$ | $-0.91_{\pm0.22}$ |
| | NegGrad+ | $-3.45_{\pm1.65}$ | $3.38_{\pm1.67}$ | $2.32_{\pm0.27}$ | $3.22_{\pm3.58}$ | $-6.99_{\pm3.53}$ | $3.75_{\pm3.53}$ |
| | NegGrad | $-19.75_{\pm0.74}$ | $19.70_{\pm0.53}$ | $0.10_{\pm0.21}$ | $-11.78_{\pm0.36}$ | $8.00_{\pm0.22}$ | $0.58_{\pm0.21}$ |
| | Random Label | $-5.20_{\pm1.78}$ | $5.13_{\pm1.81}$ | $3.22_{\pm0.67}$ | $-0.96_{\pm0.54}$ | $-2.82_{\pm0.46}$ | $3.69_{\pm0.47}$ |
| | L1-sparse | $-8.72_{\pm0.34}$ | $8.66_{\pm0.31}$ | $0.24_{\pm0.19}$ | $1.36_{\pm0.46}$ | $-2.43_{\pm0.39}$ | $1.52_{\pm0.29}$ |
| | IU | $1.57_{\pm0.28}$ | $5.04_{\pm0.91}$ | $3.11_{\pm0.18}$ | $1.45_{\pm0.36}$ | $-5.25_{\pm0.22}$ | $12.41_{\pm0.21}$ |
| **Localized Unlearning** | SSD | $2.83_{\pm1.57}$ | $-2.95_{\pm1.56}$ | $3.30_{\pm0.24}$ | $7.26_{\pm0.88}$ | $-11.09_{\pm0.85}$ | $13.26_{\pm0.74}$ |
| | CritMem-RL ($\alpha = 1\%$) | $-20.77_{\pm0.27}$ | $20.71_{\pm0.26}$ | $-0.45_{\pm0.18}$ | $-11.94_{\pm0.35}$ | $8.16_{\pm0.22}$ | $0.08_{\pm0.21}$ |
| | Shallowest-RFT ($\alpha = 30\%$) | $5.20_{\pm2.34}$ | $-5.24_{\pm2.32}$ | $3.78_{\pm1.28}$ | $3.09_{\pm0.90}$ | $-6.89_{\pm0.86}$ | $3.56_{\pm0.59}$ |
| | SalLoc-RFT ($\alpha = 30\%$) | $1.71_{\pm0.32}$ | $-1.71_{\pm0.31}$ | $2.55_{\pm0.20}$ | $0.78_{\pm0.41}$ | $4.55_{\pm0.32}$ | $3.68_{\pm0.23}$ |
| | **DEL** ($\alpha = 30\%$) | $0.75_{\pm0.91}$ | $-0.78_{\pm0.92}$ | $0.78_{\pm0.52}$ | $0.46_{\pm0.043}$ | $-4.26_{\pm0.32}$ | $0.89_{\pm0.29}$ |

## A.7 RESNET-50-IMAGENET-100 EXPERIMENTS

The results on ImageNet-100 dataset using ResNet-50 model are detailed in Table 9. Consistent with our previous results, our proposed method outperforms all compared approaches across all studied unlearning metrics. For test accuracy, our method achieves state-of-the-art performance among localized approaches; however, full-parameter methods outperform localized unlearning approaches.

Table 9: Comparison to state-of-the-art, including algorithms that update all parameters ("**Full-Parameter**") as well as ("**Localized**" unlearning algorithms on **IID** forget set when training a **ResNet-50** model on **ImageNet-100** dataset.

| | | $\Delta_{forget}$ | $\Delta_{MIA}$ | $\Delta_{test}$ |
|---|---|---|---|---|
| **Full-parameter Unlearning** | Retraining(Oracle) | $0.00_{\pm0.00}$ | $0.00_{\pm0.00}$ | $0.00_{\pm0.00}$ |
| | Fine-tuning | $-6.96_{\pm1.33}$ | $6.34_{\pm1.18}$ | $0.54_{\pm0.95}$ |
| | NegGrad+ | $-3.18_{\pm1.95}$ | $2.54_{\pm1.52}$ | $5.09_{\pm1.64}$ |
| | NegGrad | $5.13_{\pm8.05}$ | $-5.65_{\pm8.14}$ | $19.68_{\pm6.60}$ |
| | Random Label | $5.18_{\pm1.59}$ | $-5.49_{\pm1.03}$ | $5.96_{\pm1.10}$ |
| | L1-sparse | $-5.58_{\pm1.32}$ | $4.76_{\pm0.94}$ | $1.06_{\pm0.98}$ |
| **Localized Unlearning** | SSD | $-14.11_{\pm1.96}$ | $13.71_{\pm1.80}$ | $5.44_{\pm1.37}$ |
| | Shallowest-RFT ($\alpha = 30\%$) | $-1.69_{\pm2.41}$ | $2.36_{\pm2.26}$ | $11.72_{\pm1.50}$ |
| | SalLoc-RFT ($\alpha = 30\%$) | $1.36_{\pm2.01}$ | $-2.19_{\pm1.73}$ | $6.09_{\pm0.98}$ |
| | **DEL** ($\alpha = 30\%$) | $0.78_{\pm1.55}$ | $-1.74_{\pm1.35}$ | $5.20_{\pm1.08}$ |

## A.8 MEASURING UTILITY VIA RETAIN ACCURACY

Table 10 presents the retain performance when pairing various localization strategies and unlearning algorithms. The retain performance is measured as the difference between the accuracy of the oracle and unlearned models on the retain set ($\Delta_{retain} = Oracle_{retain} - Unlearn_{retain}.$). The experiments are conducted using the ResNet-18 model on the CIFAR-10 dataset with a non-IID forget set consisting of $10\%$ of randomly selected training samples. This table provides an extension of the evaluation metrics shown in Figure 3.

In terms of retain accuracy, the performance of our method is comparable to, or sometimes exceeds, the other methods of comparison. By updating only a small portion of parameters (20% or 30%) as suggested by our localization strategy, unlearning algorithms such as Random Labeling and Reset + Finetuning can achieve the Oracle retain accuracy.

Table 10: **Retain performance** ($\Delta_{retain}$) of combining different localization strategies and unlearning algorithms. The retain accuracy values from the unlearned models are provided in ($\cdot$).

| Localization Strategy | Unlearning Algorithm | | |
|---|---|---|---|
| ($\alpha$ =**parameter%**) | **Random Label** | **NegGrad+** | **Reset + Finetune** |
| CritMem($\alpha = 16\%$) | $7.36_{\pm 0.04}$ ($92.63_{\pm 0.09}$) | $0.01_{\pm 0.003}$ ($99.98_{\pm 0.005}$) | $0.02_{\pm 0.018}$ ($99.97_{\pm 0.04}$) |
| Shallowest($\alpha = 14\%$) | $7.63_{\pm 0.02}$ ($92.36_{\pm 0.04}$) | $0.02_{\pm 0.008}$ ($99.98_{\pm 0.02}$) | $0.013_{\pm 0.002}$ ($99.98_{\pm 0.004}$) |
| Shallowest($\alpha = 25\%$) | $7.41_{\pm 0.04}$ ($92.58_{\pm 0.09}$) | $0.03_{\pm 0.003}$ ($99.96_{\pm 0.04}$) | $0.007_{\pm 0.003}$ ($99.99_{\pm 0.006}$) |
| SalUn($\alpha = 20\%$) | $7.76_{\pm 0.04}$ ($92.23_{\pm 0.09}$) | $0.05_{\pm 0.005}$ ($99.94_{\pm 0.01}$) | $0.001_{\pm 0.001}$ ($99.99_{\pm 0.002}$) |
| SalUn($\alpha = 30\%$) | $6.94_{\pm 0.04}$ ($93.06_{\pm 0.04}$) | $0.05_{\pm 0.009}$ ($99.94_{\pm 0.006}$) | $0.00_{\pm 0.00}$ ($100.00_{\pm 0.00}$) |
| **Ours**($\alpha = 20\%$) | $6.97_{\pm 0.02}$ ($93.02_{\pm 0.05}$) | $-0.04_{\pm 0.007}$ ($99.95_{\pm 0.01}$) | $0.00_{0.00}$ ($100.00_{\pm 0.00}$) |
| **Ours**($\alpha = 30\%$) | $6.83_{\pm 0.02}$ ($93.16_{\pm 0.06}$) | $-0.04_{\pm 0.008}$ ($99.95_{\pm 0.02}$) | $0.00_{\pm 0.00}$ ($100.00_{\pm 0.00}$) |

