# OpenReview forum: "Improved Localized Machine Unlearning Through the Lens of Memorization"
_ICLR.cc/2025/Conference — Submitted to ICLR 2025_

### Official Review · Reviewer_EYNL · 2024-11-01

**Soundness:** 2
**Presentation:** 3
**Contribution:** 2
**Rating:** 3
**Confidence:** 3

**Summary:**

This paper addresses the challenge of machine unlearning in a localized context by introducing a novel approach based on the concept of memorization. Following a comparison of existing methods, the authors identify data-dependent and gradient-dependent techniques as particularly effective. They refine the current criticality-based localization strategy, resulting in a new unlearning algorithm, “Deletion by Example Localization” (DEL). DEL enables localized unlearning by resetting and fine-tuning parameters identified as essential based on the calculated criticality of the parameters.

**Strengths:**

1.	The paper is well-written, with a clear and concise logical flow. It begins by introducing localization as a preferred approach for model unlearning and then presents a cohesive and insightful perspective—that unlearning can be viewed as an extreme form of no memorization (lines 165-169)—which lends coherence and unity to their proposed method.
2.	The paper provides a comprehensive review of existing methods, thoroughly examining current approaches and establishing its own assumptions, such as the advantages of data-dependent over data-agnostic methods and the reasoning for utilizing gradient information. These insights serve as the foundation for their proposed method, DEL.
3.	The proposed method is both simple and effective, achieving state-of-the-art performance.

**Weaknesses:**

1. This paper extensively discusses related work and motivations, primarily focusing on comparisons between existing methods. The proposed approach appears to be a straightforward combination of existing techniques, which may limit its novelty.
2. The results in Section 3 do not necessarily support the hypotheses in Section 5.1, as the observed improvements could be attributed to other factors. Thus, a more thorough theoretical explanation of the proposed method is needed.
3. This paper focuses exclusively on classification models, but I believe that “unlearning” in LLMs (i.e., model or knowledge editing) is a more pressing concern. It remains uncertain whether the conclusions drawn from vision classifiers in this paper can be directly applied to LLMs.
4. There are a few typos, although they don’t impact comprehension. For instance, in line 159, “$f(; \theta)$” might be intended as “$f(x; \theta)$.”

**Questions:**

In Section 5.1, your paper presents several hypotheses. Could you provide a more detailed explanation of how your results support these hypotheses?

---

> ### Author Response · Authors · 2024-11-19
> **Response to Reviewer EYNL**
>
> We would like to thank the reviewer for the insightful feedback! We address the weaknesses identified by the reviewer comprehensively below, and would really appreciate hearing the reviewer’s thoughts on our responses.
>
> - **Response to W1 on limited novelty**.  We would like to respectfully push back on this. To the best of our knowledge, we are the first to investigate the link between methods that aim to localize memorization and unlearning algorithms; an investigation which led to proposing a new SOTA algorithm. We believe that this investigation is, therefore, novel and of significant scientific value (both in its own right and thanks to the resulting discovery of a new SOTA method). Further, while our method does build on existing building blocks (this is by design, and we actually view it as an advantage), its novelty arises from a carefully-selected combination of ingredients, based on our insights that are grounded in extensive empirical investigation. And, we show that it yields SOTA results across metrics, forget sets, datasets, and architectures and against both localized and full-parameter unlearning methods. At the same time, it is more robust than prior localized unlearning work to the parameter budget and outperforms prior work when paired with different unlearning algorithms, showing its versatility.  Please refer to the summary in the “common response” for an overview.
>
> - **Response to W2 and Q1: “The results in Section 3 do not necessarily support the hypotheses in Section 5.1”**. Since we have not included any results in Section 3, we kindly ask the reviewer to elaborate further on this point: which results and factors is the reviewer referring to?
> If the reviewer is referring to the results in Table 1, those do support the arguments presented in Section 5.1. Examining different granularities (parameter, channel) and criticality criteria (gradients, weighted gradients) and their impact on unlearning and utility metrics serves as an ablation study to identify the granularity and the criticality criteria that lead to a more effective unlearning algorithm. According to Table 1. (where the shaded cell represents the strategy employed in our proposed method), we conclude that using channel-wise granularity and weighted gradients is the best choice based on which we build our localization strategy.
>
> - **Response to W3: “Unlearning in LLMs is a more pressing concern”**. We agree that unlearning in LLMs is an important research area. However, unlearning in vision classifiers is also a very important and relevant research problem (e.g. for supporting deletion of user data from vision classifiers) that is largely unsolved, is equally important as its LLM counterpart, and presents different properties compared to LLM unlearning (we are happy to elaborate on this much more if the reviewer finds this discussion interesting or helpful; see e.g. [1] for some discussion). Given the fact that LLM unlearning and unlearning in vision classifiers are fundamentally different (in terms of specification of goals, metrics, and state-of-the-art methods), and we need to make progress in both, we don't think it's fair to deduct "significance" points from our work for not addressing LLM unlearning. With that said, our results with ViT demonstrate that, interestingly, our method outperforms previous SOTA in transformer-based architectures, too, making it a great candidate for future explorations in the LLM space.
>
> - **Response to W4 on typos**. Thank you for pointing this out. We have revised our draft accordingly.
>
> We look forward to hearing back from the reviewer. We would be happy to continue these conversations and address any remaining concerns that the reviewer may have.
>
> [1] Yao, Yuanshun, Xiaojun Xu, and Yang Liu. "Large language model unlearning." ICLR (2024).

---

### Official Review · Reviewer_XsVS · 2024-11-03

**Soundness:** 3
**Presentation:** 3
**Contribution:** 3
**Rating:** 6
**Confidence:** 3

**Summary:**

The paper proposes the local unlearning algorithm Deletion by Example Localization, leveraging the memorization issue. The proposed algorithm first resets the parameters that are most critical based on the localization strategy and then finetunes them. The algorithm can be paired with various existing unlearning algorithms. The author validates experiments on different datasets with different metrics to show that the performance achieves state-of-the-art.

**Strengths:**

1. The localized unlearning is new and meaningful research area and the motivation to leverage the memorization is reasonable and insightful.

2. The experiments and findings are validated with various metrics and existing unlearning algorithms and show consistently good results.

3. The paper is well formatted and arranged so that easy to understand.

**Weaknesses:**

1. There are several mathematical definitions such as the Unlearning and Label memorization. However, I did not find close connections or logical relations between them. If necessary, I expect the author to use these definitions to derive some theorems closely based on the proposed algorithm. For example, it is difficult to see theoretically or empirically if the proposed algorithm can make distribution the same as the model trained without that data.

2. Following the above, I understand in this area, most justifications are more empirical. So, I think it's better to use some metrics that can support the definition (I.e., the same distribution as a retrained model).

**Questions:**

1. I think the memorization property can vary from model scale. So, I am wondering if this memorization and proposed algorithm is available for most models since the evidence provided is empirical findings.

---

> ### Author Response · Authors · 2024-11-19
> **Response to Reviewer XsVS (1/2)**
>
> We would like to thank the reviewer for the insightful feedback and great discussion. We respond to the points raised comprehensively below and are looking forward to engaging in further discussion on these topics.
>
> - **Response to W1 on “close connections or logical relations” between the definitions of memorization and unlearning**.
>  Thank you for raising this important point. We wrote down those definitions carefully to demonstrate that the concepts of label memorization and unlearning are tightly linked to one another. Specifically, as discussed in Section 2.2, in the paragraph “connections with unlearning”, if an example isn’t memorized at all (according to Definition 2.2), it can be considered “trivially unlearned” (according to Definition 2.1). More broadly, we discuss in that same paragraph empirical work that investigates how easy it is to unlearn forget sets of different degrees of memorization. Establishing the link between these concepts is crucial to justify why we investigate memorization localization hypotheses for the purpose of informing which part of the network to perform unlearning on. Had we not shown the connection between these concepts, it would be difficult to motivate that choice. We hope that this point clarifies the link between these two concepts, as well as our motivation for providing and discussing these definitions. Now, relating to the disconnect between definitions and metrics, we discuss this in the bullet point below.
>
> - **Response to W2 on using metrics that are more aligned with the definition**. Indeed, both memorization and unlearning are quantities that are challenging to measure in a rigorous way without requiring a lot of computation (a fact that is acknowledged in extensive prior work, e.g. see the discussion in  Triantafillou et al.). The metrics for unlearning quality that we utilize are inspired by Definition 2.1 but, indeed, unavoidably make certain simplifications. For instance, it is true that comparing the accuracy of the retrained and unlearned models does not fully capture a comparison between their distributions of outputs of those models (because the argmax of the softmax can be the same, making the unlearned and retrained models to have the same “predictions”, even though the confidences / softmax distribution may be different). To amend this issue, we also use a Membership Inference Attack (MIA) that leverages the confidence of the model in order to predict whether an example was trained on or not. This is a common way to “operationalize” computing the distance between two distributions (e.g. see Fan et al., Kurmanji et al., etc.). There is recent work by Hayes et al., Triantafillou et al., (which we cite and discuss in the “Unlearning Evaluation” paragraph in Section 2.1), that designs more sophisticated metrics for unlearning that may come closer to capturing formal definitions (and representing more complex “attacks”). These, however, require training a very large number of models from each distribution and are complex to implement, requiring various design decisions and simplifying assumptions, too. For instance, the metric in Hayes et al. assumes that distributions are Gaussian, which in various cases does not hold in practice (see Fig 8 in Hayes et al. and their discussion on the limitations of this metric). Similarly, the metric of Triantafillou et al. requires implementing various “decision rules”, that require making a number of design choices based on assumptions that may not always hold. Overall, for all these reasons, evaluation in unlearning is an open research area in and of itself. We view that research direction as being orthogonal to our contributions here, so, based on the above discussion, we build on metrics that are used in recent state-of-the-art unlearning method papers (see Fan et al. whose experimental setup we adopted, for instance; but most of the other cited works use similar metrics too). We thank the reviewer for raising this important point, and we are happy to discuss more with the reviewer as well as to reflect more of this discussion in our updated paper, acknowledging that, while the metrics we use are related to the definition, they may not reflect it fully faithfully. This holds true in all empirical evaluations in the field currently.
>
> - **Response to Q1: “memorization property can vary from model scale”**. In principle, we can compute the label memorization for any model, and the memorization localization approach of Maini et al. that we build upon is an algorithm that is agnostic to the model size. Indeed, our contributions are empirical, but we have shown SOTA results on different datasets / architectures, forget sets, and across metrics (see the “common response” for an overview). We view the versatility of our method, as evidenced by its SOTA performance across the board, as strong evidence for the significance of our findings.

---

> ### Author Response · Authors · 2024-11-19
> **Response to Reviewer XsVS (2/2)**
>
> Overall, we thank the reviewer for their time and valuable feedback, and we look forward to continuing this discussion. To reiterate, while our contributions are empirical in nature, we have made substantial progress in deepening our scientific understanding on the connections between memorization and unlearning: we are the first, to the best of our knowledge, to study whether locations where memorization is hypothesized to occur give rise to better localized unlearning strategies. This gave rise to **DEL, a new algorithm that achieves SOTA results on three datasets / architectures** (CIFAR-10 with ResNet-18, SVHN with Vi,T and ImageNet-100 with ResNet-50), **various forget sets** (IID and non-IID variations of CIFAR-10), **for two different unlearning metrics, against both localized and full-parameter unlearning methods. At the same time, DEL outperforms all previous localized unlearning methods in terms of utility metrics too, which indicates that our method preserves permissible knowledge** (see e.g. Table 2, Figure 4). In addition, **DEL is more robust to the parameter budget, outperforming the previous SOTA method SalUn across different budgets, and when paired with different unlearning algorithms** (Figure 3 and Figure 2).

---

> > ### Author Response · Authors · 2024-12-03
> >
> > Dear Reviewer XsVS,
> >
> > Thank you once again for your insightful feedback. We would greatly appreciate hearing your thoughts on our response.
> >
> > To summarize: **DEL achieves SOTA results on three datasets / architectures** (CIFAR-10 with ResNet-18, SVHN with ViT, and ImageNet-100 with ResNet-50), **various forget sets** (IID and non-IID variations), **for two different unlearning metrics, against both localized and full-parameter unlearning methods. At the same time, DEL outperforms all previous localized unlearning methods in terms of utility metrics too, which indicates that our method preserves permissible knowledge** (see e.g. Table 2, Figure 4). **In addition, DEL is more robust to the parameter budget, outperforming the previous SOTA method SalUn across different budgets, and when paired with different unlearning algorithms** (Figure 3 and Figure 2).
> >
> > We believe we have thoroughly addressed all of your concerns and have strengthened our paper substantially. Based on this, we wonder if you would consider raising your score further. If not, what are the additional concerns or weaknesses of our work preventing you from doing so?

---

### Official Review · Reviewer_RSHJ · 2024-11-04

**Soundness:** 3
**Presentation:** 3
**Contribution:** 2
**Rating:** 6
**Confidence:** 3

**Summary:**

This paper introduced Deletion by Example Localization (DEL) method, which aimed at enhancing the machine unlearning by focusing on localized, a targeted data subset in neural networks. The traditional unlearning methods are removing the influence of certain data, making the model performance worse or requiring extensive retraining. However, DEL method used a selective approach by identifying a small subset of parameters that influenced by specific data points. This method can effectively remove the memory of specified data subset while persevering the model accuracy.

**Strengths:**

This method achieves state-of-the-art unlearning performance while requiring a small modification on a subset of model parameters.

This method also minimized unnecessary parameter while preserving the model efficiency.

**Weaknesses:**

The weakness of this method is limited experiments on the public dataset, only applied on CIFAR-10 and SVHN datasets, as well as the limitation on larger models.

**Questions:**

Q1: From the Appendix A.4's algorithmn, the localization strategy is mainly from the magnitude of each weighted gradient for each mini-batch. Is the localization mask determined by each mini-batch? Is the localization mask fixed for different networks? If the mask is not accurate, does it affecting the accuracy? How sensitive is DEL to different choices of localization strategy.

Q2: Does the DEL method has any specific limitations when facing more complex or diverse data distributions?

Q3: Can DEL method adapted to other network architectures? What's the differences if it adapted to a customized network structure?

Q4: Does the performance different if using different hyper-parameters, such as learning rate, batch size, etc?

Q5:  In Table 7, the accuracy is getting better with higher percentage of parameters. Will the accuracy still getting better with 40%/50%?

---

> ### Author Response · Authors · 2024-11-19
> **Response to Reviewer RSHJ (1/2)**
>
> We would like to thank the reviewer for the insightful feedback! We respond to the reviewer’s comments below:
>
> - **Response to W1 on limited datasets and models**.
> Thanks for the comment. To address your feedback, we have added another dataset / architecture pair: ImageNet-100 with ResNet-50, described in detail in Section 6 and Table 7. This dataset and architecture are significantly larger than the previous ones we considered, and the image resolution is significantly larger compared to our previous experiments. We find that, consistent with our findings on CIFAR-10 and SVHN, DEL outperforms prior methods on all unlearning metrics, while also having better test accuracy compared to all prior localized unlearning approaches. We view these new SOTA results as a strong indication for the significance of our findings and the versatility of our method across datasets and architectures. Please refer to the common response for an overview of our results, showing SOTA behavior across the board.
>
> |         | $\mathbf{\Delta_{forget}}$     | $\mathbf{\Delta_{MIA}}$      | $\mathbf{\Delta_{test}}$       |
> |-----------------|-----------------|----------------|----------------|
> | Retraining(Oracle) | $0.00_{\pm0.00}$ | $0.00_{\pm0.00}$ |$0.00_{\pm0.00}$|
> | Fine-tuning | $-6.96_{\pm1.33}$ | $6.34_{\pm1.18}$ | $0.54_{\pm0.95}$ |
> | NegGrad+ | $-3.18_{\pm1.95}$ | $2.54_{\pm1.52}$ | $5.09_{\pm1.64}$|
> | NegGrad | $5.13_{\pm8.05}$  | $-5.65_{\pm8.14}$ | $19.68_{\pm6.60}$ |
> | Random Label | $5.18_{\pm1.59}$ | $-5.49_{\pm1.03}$ | $5.96_{\pm1.10}$ |
> | L1-sparse| $-5.58_{\pm1.32}$ | $4.76_{\pm0.94}$ |  $1.06_{\pm0.98}$ |
> | SSD | $-14.11_{\pm1.96}$ | $13.71_{\pm1.80}$ | $5.44_{\pm1.37}$ |
> | Shallowest-RFT ($ \alpha $= 30%) | $-1.69_{\pm2.41}$ | $2.36_{\pm2.26}$ | $11.72_{\pm1.50}$ |
> | SalLoc-RFT($ \alpha $= 30%) | $1.36_{\pm2.01}$ | $-2.19_{\pm1.73}$ | $6.09_{\pm0.98}$ |
> | **DEL** | $\mathbf{0.78_{\pm1.55}}$ | **$\mathbf{-1.74_{\pm1.35}}$** |**$\mathbf{5.20_{\pm1.08}}$**|
>
> - **Response to Q1.1: “Is the localization mask determined by each mini-batch?”**. No, the localization mask is determined based on the magnitude of the weighted gradients over the forget set, as discussed in Section 5. This is performed by accumulating the magnitudes of the weighted gradients across all mini-batches of the forget set (see line 4 in Algorithm 1, A.4). Specifically, this approach efficiently computes the overall gradient magnitudes for the forget set by summing the weighted gradients across the mini-batches. We hope this clarifies.
>
> - **Response to Q1.2: “ Is the localization mask fixed for different networks?”**. No, the localization mask varies across different models, as it identifies the specific subset of model parameters that need to be modified to fulfill the unlearning request, using the method described above.
>
> - **Response to Q1.3: “if the mask is not accurate, does it affect the accuracy?”**. It depends. At the extreme where the mask selects to update all parameters, if we allow a sufficient finetuning budget (and the retain set is large enough), we can obtain the highest possible accuracy. This mask may not be the most “accurate” in that it selects several additional parameters rather than only the “bare minimum” that encodes the information we wish to forget, but it can obtain the highest accuracy (at the expense of computational efficiency). However, what we are interested in is the best trade-off between computing, accuracy, and unlearning quality. We hypothesize that we can achieve better trade-offs there by selecting the smallest set of weights that we should operate on for unlearning. And we have strong evidence for this: DEL outperforms all prior methods in terms of unlearning quality across the board (and all prior localized methods in terms of accuracy) and is SOTA across various parameter budgets (see e.g. Figure 2).
>
> - **Response to Q2 on different data distibutions**. We have applied DEL on two different types of forget sets, one that is sampled IID from the training data and one that is non-IID (from a subset of classes). We find that DEL outperforms prior methods in both cases. Does this answer the reviewer’s question?
>
> - **Response to Q3 on different architectures**. Yes, We have already shown results using three different architectures: a ResNet-18 and ResNet-50, which are convolutional networks, and a ViT, which is a transformer. We believe ResNet and ViT cover the two most widely used categories of architectures, and the fact that DEL is SOTA in both cases is important evidence of its versatility.

---

> > ### Comment · Reviewer_RSHJ · 2024-12-03
> >
> > Appreciate the authors' detailed responses. Glad to see the new results for ImageNet-100 with ResNet-50. It does show the SOTA results in all of the testing dataset, which makes me comfortable to increase my score. Thanks for the author claimed that the method also works on different architectures and data distributions.

---

> > > ### Author Response · Authors · 2024-12-03
> > >
> > > We sincerely thank you for your response to our rebuttal, for acknowledging our new results on ImageNet, and for updating your score. We believe that we have fully addressed your concerns and strengthened our paper substantially.
> > >
> > > To summarize: **DEL achieves SOTA results on three datasets / architectures** (CIFAR-10 with ResNet-18, SVHN with ViT, and ImageNet-100 with ResNet-50), **various forget sets** (IID and non-IID variations), **for two different unlearning metrics, against both localized and full-parameter unlearning methods. At the same time, DEL outperforms all previous localized unlearning methods in terms of utility metrics too, which indicates that our method preserves permissible knowledge** (see e.g. Table 2, Figure 4). **In addition, DEL is more robust to the parameter budget, outperforming the previous SOTA method SalUn across different budgets, and when paired with different unlearning algorithms** (Figure 3 and Figure 2).
> > >
> > > Based on this, we wonder if you would consider raising your score further. If not, what are the additional concerns or weaknesses of our work preventing you from doing so?

---

> ### Author Response · Authors · 2024-11-19
> **Response to Reviewer RSHJ (2/2)**
>
> - **Response to Q4 on different hyper-parameters**. Yes, the performance is dependent on the hyperparameters, as is the case for any method. For fair and careful experimentation, we tuned the hyperparameters of our method and each baseline separately for each scenario and parameter budget we considered. Please see Section A.2 for full details. Note as well that we find that our localization method is quite versatile: it pairs well with different unlearning methods and is more robust to prior state-of-the-art to the parameter budget (Figure 3). We view this as an important advantage.
>
> - **Response to Q5 on Tabel 7**. Generally, it is possible that the accuracy improves for an increased budget of parameters. The best way to see this is through the rightmost subplot of Figure 2 - note that Table 7  (currently Table 10 in the revised paper) that the reviewer refers to shows the retain set accuracies, so it may be a little less relevant compared to test accuracy and other metrics. However, as mentioned in the response to Q1 as well, we are more interested in the *trade-off* between efficiency (e.g. using parameter budget as a proxy), unlearning quality and accuracy. Otherwise, one could just retrain the entire network from scratch to obtain perfect unlearning. Our rationale for considering localized unlearning is the hypothesis that it yields better such trade-offs. Indeed, through our extensive experimentation we show that DEL obtains the best unlearning quality both compared to localized *and full-parameter unlearning* methods, and that it outperforms all state-of-the-art prior localized unlearning methods in terms of accuracy too, thus introducing a new interesting point in the pareto frontier, and enhancing our scientific understanding of relevant trade-offs in unlearning methods.

---

### Official Review · Reviewer_j7JZ · 2024-11-07

**Soundness:** 3
**Presentation:** 1
**Contribution:** 3
**Rating:** 5
**Confidence:** 3

**Summary:**

This work attempted to tackle the problem of localized unlearning by investigating it based on the memorization assumption and proposed DEL for some parameters with resetting and fine-tuning. The proposed method showed promising results on forgetting on a couple of benchmarks.

**Strengths:**

- I liked the initial idea of investigating localized unlearning based on memorization.
- The proposed method was partially successful on some forgetting benchmarks.

**Weaknesses:**

- The method is based on a lot of assumptions without much justification, but with intuition. Thus, it is very hard to see if the proposed method is indeed ok in terms of unlearning (while preserving the rest!).
- It is very hard to see the core contribution clearly due to poor writing. It was very hard to read and follow.
- Experiments look quite limited in terms of benchmarks (datasets, compared methods). I am afraid that the localized unlearning approach may hurt the preservation of remaining parts, but it is unclear if it is true.

**Questions:**

- Please address the concerns in the weakness section.

---

> ### Author Response · Authors · 2024-11-19
> **Response to Reviewer j7JZ (1/2)**
>
> We would like to thank the reviewer for their time. We are frankly surprised and puzzled with this score and with the reviewer’s feedback. The reviewer makes a number of assertive statements which are factually incorrect. Please see our responses below. We would really appreciate hearing back from the reviewer on these.
>
> First, let us reiterate our contributions:
>
> A) We perform the first, to the best of our knowledge, study of whether hypotheses for where memorization occurs in a network can give rise to improved localized unlearning algorithms, through informing which subset of parameters to modify during unlearning. This is the first attempt that we are aware of to bridge memorization localization methods with unlearning algorithms. Our analysis revealed previously-unknown trade-offs between different data-agnostic and data-dependent localization strategies on several metrics of interest (unlearning quality and utility metrics) and under different parameter budgets (Figure 2).
>
> B) Building on those insights and on extensive empirical investigations,  we propose a new localized unlearning method, DEL, that borrows the deemed-to-be most successful “ingredients” (criticality criterion, granularity of localization) from the memorization literature and ports them into a framework that yields an efficient and practical localization algorithm for unlearning.
>
> C) Overall, we find that **DEL achieves SOTA results on three datasets / architectures** (CIFAR-10 with ResNet-18, SVHN with ViT and ImageNet-100 with ResNet-50), **various forget sets** (IID and non-IID variations of CIFAR-10), **for two different unlearning metrics, against both localized and full-parameter unlearning methods. At the same time, DEL outperforms all previous localized unlearning methods in terms of utility metrics, too, which indicates that our method preserves permissible knowledge** (see, e.g., Table 2, Figure 4). In addition, **DEL is more robust to the parameter budget, outperforming the previous SOTA method SalUn across different budgets, and when paired with different unlearning algorithms** (Figure 3 and Figure 2).
>
> Overall, we view these contributions as a significant step forward in the development of localized unlearning methods, as well as growing our scientific understanding of behaviors and trade-offs of different memorization localization hypotheses for the purpose of unlearning.
>
> Now, to address the reviewer’s comments specifically:
>
> - **Response to W1.1: “based on a lot of assumptions without justification”**. Our method does not make additional assumptions that we are aware of compared to prior work. What does the reviewer have in mind here? Further, it is not true that our method lacks justification. It is based on hypotheses that are investigated via extensive experiments (see Figure 2 and Table 1) – for instance, as reviewer EYNL put it, ”These insights serve as the foundation for their proposed method, DEL.”. We respectfully argue that empirical results of carefully-designed experiments do qualify as “justification” and that our method is well-grounded in our findings and insights.
>
> - **Response to W1.2: “it is very hard to see if the proposed method is indeed ok in terms of unlearning (while preserving the rest!)”**.
>  We are puzzled by this comment. We have evaluated our method comprehensively on different forget sets, datasets, and architectures, and different metrics (two metrics for forgetting quality as well as utility metrics) against SOTA methods for both localized as well as full-parameter unlearning. We find that our method outperforms the previous SOTA across the board. This is evidence both that it is “ok in terms of unlearning” and that it “preserves the rest” (which is captured through the utility metrics).
>
> - **Response to W2: “It is very hard to see the core contribution clearly due to poor writing. It was very hard to read and follow”**. We are very surprised to see this comment. We put a lot of care into writing our paper and all other reviewers found it well written. Does the reviewer have any specific feedback or suggestions for how we should improve our writing?

---

> ### Author Response · Authors · 2024-11-19
> **Response to Reviewer j7JZ (/2)**
>
> - **Response to W3.1: “Experiments look quite limited in terms of benchmarks (datasets, compared methods)”**. During the rebuttal, we have added experiments using an additional dataset and architecture pair: ImageNet-100 with ResNet-50, described in detail in Section A.6 and Table 8 in the revised paper. This dataset and architecture are significantly larger than the previous ones we considered, and the image resolution is significantly larger compared to our previous experiments. We find that, consistent with our findings on CIFAR-10 and SVHN, DEL outperforms prior methods on all unlearning metrics, while also having better test accuracy compared to all prior localized unlearning approaches. We view these new SOTA results as an additional strong indication for the significance of our findings and the versatility of our method across datasets and architectures. Please refer to the common response for an overview of our results, showing SOTA behavior across the board. Regarding “compared methods”, we have compared against all the SOTA methods we are aware of, and during the rebuttal we additionally added a comparison to the contemporaneous work of Foster et al. (SSD) as well as the influence unlearning method of Izzo et al. (IU) . Please refer to the updated Table 2 and Table 8 in the revised paper. We find that DEL outperforms all prior methods.
>
> - CIFAR10-ResNet-18 with IID forget set
>
> |         | $\mathbf{\Delta_{forget}}$     | $\mathbf{\Delta_{MIA}}$      | $\mathbf{\Delta_{test}}$       |
> |-----------------|-----------------|----------------|----------------|
> | Retraining(Oracle) | $0.00_{\pm0.00}$ | $0.00_{\pm0.00}$ |$0.00_{\pm0.00}$|
> | IU | $-2.20_{\pm0.39}$ | $2.19_{\pm0.38}$ | $10.94_{\pm0.43}$|
> | SSD | $1.60_{\pm1.99}$ | $1.59_{\pm1.98}$ | $11.58_{\pm1.03}$ |
> | **DEL** | $\mathbf{0.97_{\pm0.42}}$ | **$\mathbf{-0.97_{\pm0.40}}$** |**$\mathbf{1.87_{\pm0.49}}$**|
>
> - CIFAR10-ResNet-18 with non-IID forget set
>
> |         | $\mathbf{\Delta_{forget}}$     | $\mathbf{\Delta_{MIA}}$      | $\mathbf{\Delta_{test}}$       |
> |-----------------|-----------------|----------------|----------------|
> | Retraining(Oracle) | $0.00_{\pm0.00}$ | $0.00_{\pm0.00}$ |$0.00_{\pm0.00}$|
> | IU | $-5.00_{\pm0.88}$ | $5.04_{\pm0.91}$ | $4.18_{\pm0.19}$ |
> | SSD | $-11.16_{\pm6.28}$ | $11.18_{\pm6.29}$ | $2.68_{\pm1.18}$ |
> | **DEL** | $\mathbf{0.43_{\pm1.06}}$ | **$\mathbf{0.64_{\pm1.23}}$** |**$\mathbf{2.23_{\pm0.25}}$**|
>
> - SVHN-ViT with IID forget set
>
> |         | $\mathbf{\Delta_{forget}}$     | $\mathbf{\Delta_{MIA}}$      | $\mathbf{\Delta_{test}}$       |
> |-----------------|-----------------|----------------|----------------|
> | Retraining(Oracle) | $0.00_{\pm0.00}$ | $0.00_{\pm0.00}$ |$0.00_{\pm0.00}$|
> | IU | $1.45_{\pm0.36}$ | $-5.25_{\pm0.22}$ | $12.41_{\pm0.21}$ |
> | SSD | $7.26_{\pm0.88}$ | $-11.09_{\pm0.85}$ | $13.26_{\pm0.74}$|
> | **DEL** | $\mathbf{0.46_{\pm0.043}}$ | **$\mathbf{-4.26_{\pm0.32}}$** |**$\mathbf{0.89_{\pm0.29}}$**|
>
> - SVHN-ViT with non-IID forget set
>
> |         | $\mathbf{\Delta_{forget}}$     | $\mathbf{\Delta_{MIA}}$      | $\mathbf{\Delta_{test}}$       |
> |-----------------|-----------------|----------------|----------------|
> | Retraining(Oracle) | $0.00_{\pm0.00}$ | $0.00_{\pm0.00}$ |$0.00_{\pm0.00}$|
> | IU | $1.57_{\pm0.28}$ | $5.04_{\pm0.91}$ | $3.11_{\pm0.18}$ |
> | SSD | $2.83_{\pm1.57}$ | $-2.95_{\pm1.56}$ | $3.30_{\pm0.24}$ |
> | **DEL** | $\mathbf{0.75_{\pm0.91}}$ | **$\mathbf{-0.78_{\pm0.92}}$** |**$\mathbf{0.78_{\pm0.52}}$**|
>
> - **Response to W3.2: “I am afraid that the localized unlearning approach may hurt the preservation of remaining parts, but it is unclear if it is true.”**.  As mentioned above, this is exactly what the utility metrics capture (specifically, these are the accuracy of the unlearned model on the retain set and the test set). And through our thorough investigation, we find that our method outperforms the previous state-of-the-art localized unlearning in terms of these metrics.
>
> In summary, the reviewer’s feedback seems to dismiss our empirical results, which provide a solid grounding for our novel method, and the ample evidence that DEL yields SOTA performance both in terms of unlearning metrics and utility. The harsh tone of the review and associated score are not corroborated by factual criticism with concrete reference to specific parts of our work. We are nonetheless keen to engage in a grounded scientific discussion about our work. We believe that the extensive clarifications above and the additional results on another dataset / architecture and additional baseline have addressed all criticism. We look forward to hearing from the reviewer if their stance is changed based on this. If not, we would like to know in what ways, in the reviewer’s opinion, the paper can be concretely improved.

---

> ### Comment · Reviewer_j7JZ · 2024-12-02
>
> Dear the authors,
>
> I appreciate your detailed comments - many of my concerns were lifted, so I increased my score to 5 (from 1). I must say that my initial score was quite low due to my misunderstanding for the overall structure of this work and your rebuttal clearly elaborated them (for some reason, I still have a hard time to read the revision, though).
>
> This work was based on the memorization assumption, but I was not able to see 'any' memorization related metrics or evidence in the experiments. I meant 'without justification' by that. Even though the overall performance look good, it is still unclear if it was from the intended assumption on leveraging memorization or somewhere else.
>
> I am glad to see the new results for ImageNet, which helped me a lot to alleviate my major concern (so I was comfortable to increase my score by a large margin). Initially, I was not quite sure if the memorization can indeed happen for small network with small benchmarks, but ImageNet results look promising and I am a little bit more convinced.
>
> Lastly, it could be better if the proposed method was compared with some of the recent state-of-the-art methods such as SCRUB, but the current comparisons could be enough.
>
>
> See the following recent work, which may be related to the current work:
> AK Tarun et al., Fast Yet Effective Machine Unlearning, IEEE Trans Neural Networks and Learning Systems 3(9), Sep 2024.

---

> > ### Author Response · Authors · 2024-12-03
> >
> > Thank you for your response, we really appreciate the additional discussion. Please find our responses below, which we believe have addressed your original and new concerns in depth and warrant a further increase in your score.
> >
> > - **Response to :  “for some reason, I still have a hard time to read the revision, though”**- please let us know what remains unclear. We are very committed to improving our work based on your feedback.
> >
> > - **Response to : “I was not able to see ‘any’ memorization related metrics”** - this is a great point, please let us further clarify. As per the discussion in Section 2.2, memorization and unlearning and very tightly connected concepts (based on Definitions 2.1 and 2.2). Consequently, memorization-like metrics (like membership inference attacks, e.g. see Jagielski et al. 2022) are in fact what is commonly used for evaluating unlearning (see e.g. Fan et al, Hayes et al, for membership inference attack-based unlearning metrics). Intuitively, a successful unlearning algorithm alleviates the memorization of the examples in the forget set. In our paper, we will clarify the tight connection between the metrics used to measure memorization and the metrics used to measure unlearning quality. Thank you for this comment.
> >
> > - **Response to : factors behind our strong performance**. In addition to the above discussion on the connections between memorization and unlearning, we wanted to bring to the reviewer’s attention that we have conducted substantial ablation studies (for different criticality criteria, etc). We also study whether the selection of critical parameters is responsible for good performance, in Section 6 (Table 3).
> >
> > - **Response to : comparison with SCRUB.** We can add this additional method in the revision too, but we omitted it because 1) it is outperformed by the latest SOTA, which we have already compared against, and 2) as documented in the SCRUB paper itself, it performs very similarly to NegGrad+ (which we have included in our comparisons) in terms of their behaviors, 3) it is complex to tune SCRUB’s hyperparameters, so given 1 and 2, we decided against implementing this method.  Because of 1, 2 and 3, we don’t believe that this omission affects the validity of our findings in any way.
> >
> > - **Response to :  “Fast yet effective machine unlearning”** - thank you for bringing this paper to our attention, we will include it in the revised paper. However, that work focuses on class unlearning, whereas we study a related but different problem: unlearning memorized data, which may or may not all belong to the same class. Our goal is to match the distribution of predictions made by retraining from scratch without the forget set, which is different from the goal of Tarun et al. We will explain these differences in our revised related work section.
> >
> > Thank you for acknowledging our new results on ImageNet, we agree that this large-scale dataset and architecture indeed is strong evidence for the versatility and generality of our method.
> >
> > Overall, we are very grateful for the reviewer’s response and additional discussion. We politely argue that, based on the above additional clarifications that we believe address all initial and additional concerns of the reviewer, our paper deserves a higher score than a 5.
> >
> > As a summary of our contributions: **DEL achieves SOTA results on three datasets / architectures** (CIFAR-10 with ResNet-18, SVHN with ViT, and ImageNet-100 with ResNet-50), **various forget sets** (IID and non-IID variations of CIFAR-10), **for two different unlearning metrics, against both localized and full-parameter unlearning methods. At the same time, DEL outperforms all previous localized unlearning methods in terms of utility metrics too, which indicates that our method preserves permissible knowledge** (see e.g. Table 2, Figure 4). **In addition, DEL is more robust to the parameter budget, outperforming the previous SOTA method SalUn across different budgets, and when paired with different unlearning algorithms** (Figure 3 and Figure 2).

---

### Author Response · Authors · 2024-11-19
**General response to all reviewers**

Dear reviewers,

We thank you for your time and valuable feedback! We have worked hard during the rebuttal, and we believe we have addressed all of the concerns comprehensively. We look forward to hearing back from the reviewers and discussing further.

We address each reviewer’s feedback individually in separate comments. In this common response, we will present new results that we ran during the rebuttal to address reviewers’ feedback.

A) We have implemented and compared against SSD of Foster et al. as an additional baseline, which we excluded from the original submission as it is contemporaneous work. Additionally, we include comparisons with the Influence Unlearning (IU) method proposed by Izzo et al. See the updated Table 2 and Table 8 in the revised paper. We find that DEL significantly outperforms these baselines too, on both types of forget sets and on all metrics considered.

B) We additionally conducted experiments on a new dataset, a subset of ImageNet (see the new Section A.6 and the results in Table 8 in the revised paper) using a ResNet-50 architecture. This dataset and architecture are significantly larger than the previous ones we considered, and the image resolution is significantly larger compared to our previous experiments. We find that, consistent with our findings on CIFAR-10 and SVHN, DEL outperforms prior methods on all unlearning metrics while also having better test accuracy compared to all prior localized unlearning approaches.

Overall, we find that **DEL achieves SOTA results on three datasets / architectures** (CIFAR-10 with ResNet-18, SVHN with ViT, and ImageNet-100 with ResNet-50), **various forget sets** (IID and non-IID variations of CIFAR-10), **for two different unlearning metrics, against both localized and full-parameter unlearning methods. At the same time, DEL outperforms all previous localized unlearning methods in terms of utility metrics too, which indicates that our method preserves permissible knowledge** (see e.g. Table 2, Figure 4). In addition, **DEL is more robust to the parameter budget, outperforming the previous SOTA method SalUn across different budgets, and when paired with different unlearning algorithms** (Figure 3 and Figure 2).

---

### Author Response · Authors · 2024-11-24
**Looking forward to feedback on our rebuttal**

Dear reviewers,

We have worked very hard during the rebuttal and we believe we have addressed all of your concerns. We would really appreciate hearing back from you about our responses.

---

### Comment · Reviewer_EYNL · 2024-12-03
**My final rating is 5: marginally below the acceptance threshold**

I understand the author's responses and efforts. I  change my rating into 5: marginally below the acceptance threshold.

---

> ### Author Response · Authors · 2024-12-03
>
> We truly thank the reviewer for their response to our rebuttal and for updating the score.
>
> We kindly request that you also update the score reflected in your written review, as it currently shows the previous rating. We believe you can do this on OpenReview by selecting 'edit' on your original review.
>
> Overall, we sincerely thank you for your time and valuable feedback. We believe that we have fully addressed your concerns and strengthened our paper substantially.
>
> To summarize: **DEL achieves SOTA results on three datasets / architectures** (CIFAR-10 with ResNet-18, SVHN with ViT, and ImageNet-100 with ResNet-50), **various forget sets** (IID and non-IID variations), **for two different unlearning metrics, against both localized and full-parameter unlearning methods. At the same time, DEL outperforms all previous localized unlearning methods in terms of utility metrics too, which indicates that our method preserves permissible knowledge** (see e.g. Table 2, Figure 4). **In addition, DEL is more robust to the parameter budget, outperforming the previous SOTA method SalUn across different budgets, and when paired with different unlearning algorithms** (Figure 3 and Figure 2).
>
> Based on this, we wonder if you would consider raising your score further. If not, what are the additional concerns or weaknesses of our work preventing you from doing so?

---

### Meta-Review · Area_Chair_4KVV · 2024-12-21

**Metareview:**

Summary. This paper introduced Deletion by Example Localization (DEL) method, which aimed at enhancing the machine unlearning by focusing on localized, a targeted data subset in neural networks. This method can effectively remove the memory of specified data subset while persevering the model accuracy.

Strengths.
The idea of localized unlearning is interesting.
The paper provides a detailed background to motivate the problem and a review of existing methods.

Weaknesses.
Parts of the unlearning process are unclear, which makes it difficult to understand the core contributions of the paper. While the paper clearly describes several localization and unlearning strategies, it is unclear what are the proposed strategies, are they novel, or are they a combination of known techniques?
The experiments are presented on small datasets (CIFAR-10 with ResNet-18 and SVHN with ViT; ImageNet-100 with ResNet-50 added in rebuttal). These experiments seem limited in scope and impact compared to other unlearning papers.
The paper focuses exclusively on the classification models, which is a limitation at this stage.

Missing.
A clear description of the algorithm with an emphasis on novel aspects will be useful.
Experiments on models beyond classification can provide useful insights in the generalization of the proposed method.

Reasons.
The technical novelty of the proposed work is unclear. Localized unlearning methods exist, and this paper offers some modifications for the improvement. Experiments are mainly performed on classification models, which seems limited in scope and impact.

**Additional Comments On Reviewer Discussion:**

The paper was discussed among authors and reviewers.

Reviewers raised concerns about clarity, novelty, and limited experiments.

Authors responded to the comments; offered clarifications and provided an additional experiment on ImageNet-100 dataset.

Overall, the reviewers have mixed ratings (5 and 6). I weighted all the reviewer comments equally and lean toward reject.

---

### Decision · Program_Chairs · 2025-01-22

Reject